Letter

# The wheat stem rust resistance gene *Sr43* encodes an unusual protein kinase

Guotai Yu[1,2,3], Oadi Matny [4], Spyridon Gourdoupis [5], Naganand Rayapuram [1,2], Fatimah R. Aljedaani [1,2], Yan L. Wang[6], Thorsten Nürnberger[6], Ryan Johnson[4], Emma E. Crean [7], Isabel M.-L. Saur [7,8], Catherine Gardener[1,2,3], Yajuan Yue[3], Ngonidzashe Kangara[3], Burkhard Steuernagel [3], Sadiye Hayta[3], Mark Smedley[3], Wendy Harwood [3], Mehran Patpour[9], Shuangye Wu[10], Jesse Poland [1,2,10], Jonathan D. G. Jones [11], T. Lynne Reuber[12,19], Moshe Ronen[13], Amir Sharon [14], Matthew N. Rouse[4,15], Steven Xu[16], Kateřina Holušová[17], Jan Bartoš [17], István Molnár [17,20], Miroslava Karafiátová [17], Heribert Hirt[1,2], Ikram Blilou[1,2], Łukasz Jaremko[5,18], Jaroslav Doležel [17], Brian J. Steffenson[4] ✉ & Brande B. H. Wulff [1,2,3] ✉

To safeguard bread wheat against pests and diseases, breeders have introduced over 200 resistance genes into its genome, thus nearly doubling the number of designated resistance genes in the wheat gene pool[1]. Isolating these genes facilitates their fast-tracking in breeding programs and incorporation into polygene stacks for more durable resistance. We cloned the stem rust resistance gene *Sr43*, which was crossed into bread wheat from the wild grass *Thinopyrum elongatum*[2,3]. *Sr43* encodes an active protein kinase fused to two domains of unknown function. The gene, which is unique to the Triticeae, appears to have arisen through a gene fusion event 6.7 to 11.6 million years ago. Transgenic expression of *Sr43* in wheat conferred high levels of resistance to a wide range of isolates of the pathogen causing stem rust, highlighting the potential value of *Sr43* in resistance breeding and engineering.

Worldwide, ~20% of projected bread wheat (*Triticum aestivum*) production is lost to pests and diseases every year[4]. The deployment of genetic variation for disease resistance is a sustainable and environmentally friendly way to protect wheat crops[5]. For over 100 years, breeders have conducted numerous crosses to enrich the wheat gene pool with resistance genes. Notably, more than 200 of the 467 currently designated resistance genes in cultivated bread wheat have their origin outside the bread wheat gene pool[1]. However, the deployment of these interspecific

[1]Plant Science Program, Biological and Environmental Science and Engineering Division (BESE), King Abdullah University of Science and Technology (KAUST), Thuwal, Saudi Arabia. [2]Center for Desert Agriculture, KAUST, Thuwal, Saudi Arabia. [3]John Innes Centre, Norwich Research Park, Norwich, UK. [4]Department of Plant Pathology, University of Minnesota, St. Paul, MN, USA. [5]Bioscience Program, Smart Health Initiative, BESE, KAUST, Thuwal, Saudi Arabia. [6]Department of Plant Biochemistry, Centre of Plant Molecular Biology (ZMBP), University of Tübingen, Tübingen, Germany. [7]Institute for Plant Sciences, University of Cologne, Cologne, Germany. [8]Cluster of Excellence on Plant Sciences (CEPLAS), Cologne, Germany. [9]Department of Agroecology, Aarhus University, Slagelse, Denmark. [10]Department of Plant Pathology, Kansas State University, Manhattan, KS, USA. [11]The Sainsbury Laboratory, University of East Anglia, Norwich, UK. [12]2Blades Foundation, Evanston, IL, USA. [13]Institute for Cereal Crops Research, Tel Aviv University, Tel Aviv, Israel. [14]Institute for Cereal Crops Research, and the School of Plant Sciences and Food Security, Tel Aviv University, Tel Aviv, Israel. [15]USDA-ARS, Cereal Disease Laboratory, St. Paul, MN, USA. [16]Crop Improvement and Genetics Research Unit, USDA-ARS, Western Regional Research Center, Albany, CA, USA. [17]Centre of the Region Haná for Biotechnological and Agricultural Research, Institute of Experimental Botany of the Czech Academy of Sciences, Olomouc, Czech Republic. [18]Red Sea Research Center, BESE, KAUST, Thuwal, Saudi Arabia. [19]Present address: Enko Chem, Mystic, CT, USA. [20]Present address: Centre for Agricultural Research, ELKH, Agricultural Institute, Martonvásár, Hungary. ✉e-mail: bsteffen@umn.edu; brande.wulff@kaust.edu.sa

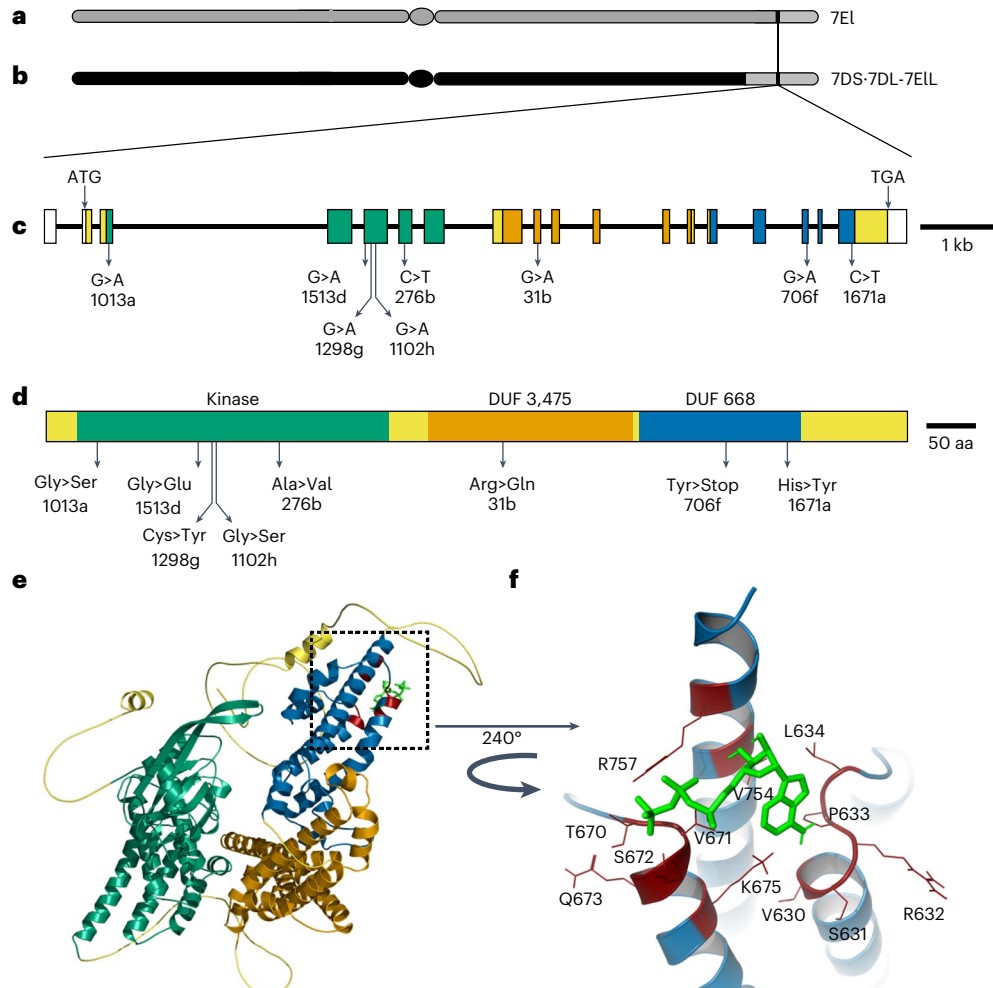

**Fig. 1 | Sr43 encodes a protein kinase fused to two DUFs. a**, *Th. elongatum* chromosome 7. **b**, Schematic diagram of the wheat–*Thinopyrum* translocation chromosome. **c**, Identified EMS mutations along the predicted *Sr43* gene model. **d**, Schematic diagram of Sr43, showing predicted domains and amino acid changes induced by the EMS mutations. **e**, Three-dimensional model of Sr43, as predicted by AlphaFold. Green, kinase; orange and blue, DUF domains; yellow, flexible linkers. **f**, Structural detail (dashed box in **e**) of a high-confidence ATP-binding site (red residues) in DUF668 bound to an ATP molecule, as determined by the small molecule docking program HADDOCK[28]. ATP is depicted as a stick structure (light green) connected to DUF668 residues by hydrogen bonds (red lines).

resistance genes is often hampered by linkage drag, that is, the cointroduction of deleterious alleles from linked genes. Moreover, single resistance genes tend to be rapidly overcome by the emergence of resistance-breaking pathogen strains[6]. Cloning individual resistance genes would enable their introduction as genetically modified polygene stacks, which are likely to provide more durable resistance[7].

Most of the ~291 plant disease resistance genes cloned to date encode either intracellular receptors of the nucleotide-binding and leucine-rich repeat (NLR) class or extracellular membrane-anchored receptor-like proteins (RLPs, called RLKs when they contain an intracellular kinase) (Supplementary Table 1)[1,8]. A new group of resistance genes has recently come to light, whose members encode two protein kinases fused as one protein. These tandem kinase genes include *Rpg1*, *Yr15*, *Sr60*, *Sr62*, *Pm24*, *WTK4* and *Rwt4* (refs. 9–15). Other resistance genes offer some variation to this architecture with protein kinases fused to a steroidogenic acute regulatory protein-related lipid transfer domain (*Yr36*)[16], a C2 domain and a multitransmembrane region (*Pm4*)[17], a major sperm protein (*Snn3*)[18], an NLR (*Tsn1*, *Rpg5* and *Sm1*)[19–21] and a von Willebrand factor type A domain (*Lr9*)[22].

These kinase fusion protein-encoding resistance genes appear to be unique to the Triticeae, the clade of grasses that arose 12 million years

ago and encompasses the cereals wheat, rye (*Secale cereale*) and barley (*Hordeum vulgare*)[23]. However, the fusion events that gave rise to these genes, far from being rare and isolated, happened many times between different classes of kinases and spawned diverse combinations[10,12]. This genomic innovation resulted in resistance against phylogenetically distinct fungal pathogens spanning the ~300 million-year-old ascomycete/basidiomycete divide.

Here, we cloned the stem rust resistance gene *Sr43*, which was transferred from tall wheatgrass (*Thinopyrum elongatum*) into bread wheat 45 years ago[2,3]. The dominant resistance gene *Sr43* was introgressed into chromosome 7D of hexaploid wheat (Fig. 1a,b). We mutagenized grains of the *Sr43* introgression line with ethyl methanesulfonate (EMS) and screened 2,244 surviving M$_2$ families for susceptibility to *Puccinia graminis* f. sp. *tritici* (*Pgt*). We identified 23 families segregating for stem rust susceptibility, of which we confirmed ten independent mutants by progeny testing (Supplementary Table 2) and genotyping (Supplementary Figs. 1 to 11).

To clone *Sr43*, we performed chromosome flow sorting and sequenced the wheat–*Th. elongatum* recombinant chromosome 7D in the parental line and eight mutants (Extended Data Fig. 1 and Supplementary Tables 3 and 4). Sequence assembly of the parental

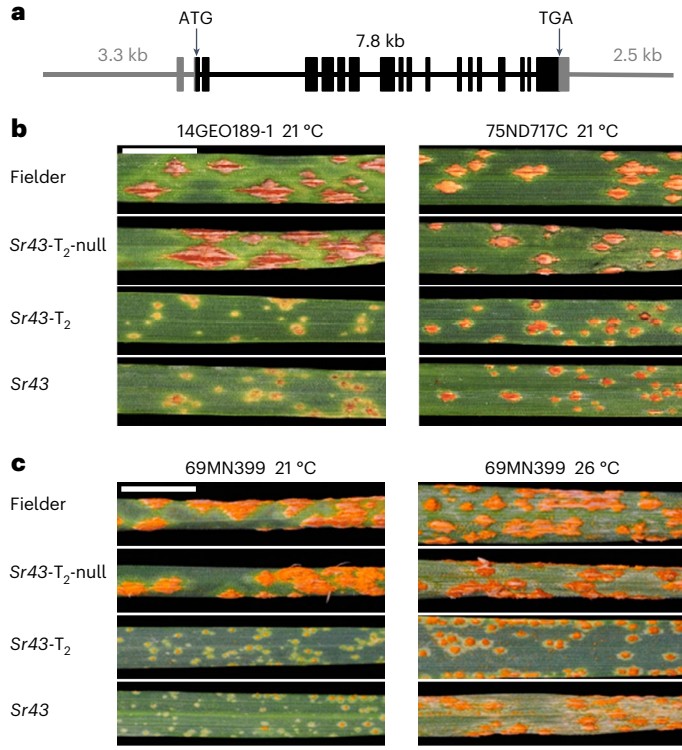

**Fig. 2 | Confirmation of *Sr43* function, race specificity and temperature sensitivity. a**, Schematic diagram of the *Sr43* genomic fragment used for transformation into wheat cv. Fielder. **b**, Representative leaves from seedlings of *Sr43* wild-type and transgenic lines alongside non-transgenic wild-type Fielder and null controls inoculated with *P. graminis* f. sp. *tritici* isolates 14GEO189-1 (avirulent on *Sr43*) and 75ND717C (intermediately virulent on *Sr43*). **c**, Effect of temperature on *Sr43*-mediated resistance to *Pgt* isolate 69MN399. Scale bar, 1 cm.

line and mapping of the mutant reads identified a 10 kilobase (kb) window in a scaffold containing a mutation in all eight mutants. To determine the gene structure of the *Sr43* candidate, we (1) conducted transcriptome deep sequencing (RNA-seq) analysis of *Sr43* seedling leaves and mapped the reads to the *Sr43* genomic scaffold (Extended Data Fig. 2a) and (2) sequenced *Sr43* clones obtained by polymerase chain reaction (PCR) from a full-length complementary DNA library. We detected four different splice variants (Extended Data Fig. 2b and Supplementary Tables 5 and 6). Splice variant 1 contained all eight mutations and consisted of 18 exons with a predicted open reading frame of 2,598 base pairs (bp) (Fig. 1c). The eight mutations were all G/C to A/T transitions typical of EMS mutagenesis and introduced non-synonymous changes (seven mutants) or an early stop codon in the predicted coding sequence (Fig. 1c,d and Supplementary Tables 7 and 8). The probability that all mutants would have a mutation in the same gene by chance alone out of the 5,822 non-redundant genes of chromosome 7D (ref. 24) was $4 \times 10^{-6}$, indicating that the identified gene is a good candidate for *Sr43*.

As all identified EMS mutations affected the predominant full-length *Sr43* transcript (Fig. 1c), we used its predicted 866–amino acid sequence to search for functional domains and homologs. We determined that Sr43 harbors an N-terminal kinase domain and two domains of unknown function (DUFs) in its C terminus (Fig. 1d). Five of the mutations resided within the kinase domain, with the remaining three mutations affecting either DUF (Fig. 1d).

The closest BLAST homolog of the Sr43 kinase domain was the serine/threonine kinase interleukin-1 receptor-associated kinase (STKc IRAK) (Supplementary Fig. 12), indicating that Sr43 is probably a kinase. Further homology searches suggested that the kinase domain is intact (Supplementary Fig. 13). Supporting this observation, we found that an

affinity-tagged Sr43 fusion protein purified from *Eschericia coli* phosphorylated maltose-binding protein DNA gyrase in vitro (Supplementary Figs. 14–16 and Supplementary Table 9). Moreover, mutant 1013a disrupted one of the conserved glycine residues in the glycine-rich loop, suggesting that kinase activity is required for Sr43 function (Supplementary Table 10). The C terminus of Sr43, containing DUF3475 and DUF668, has a similar domain architecture (44% identity) to the N terminus of PHYTOSULFOKINE SIMULATOR (PSI) proteins from Arabidopsis (*Arabidopsis thaliana*), which are critical for plant growth[25]. Unlike Arabidopsis PSI1, Sr43 lacked a putative nuclear localization signal or a putative myristoylation site. Sr43 had no transmembrane domain, as predicted by InterPro. However, we established that Sr43 probably localizes to the nucleus, cytoplasm and plastids, as evidenced by the fluorescence detected from the transient expression of a *Sr43-GFP* (green fluorescent protein) construct in *Nicotiana benthamiana* leaf epidermal cells (Extended Data Fig. 3). The nuclear and cytoplasmic localization was confirmed in wheat protoplasts transfected with Sr43-GFP (Extended Data Fig. 3).

The domain structure of Sr43 was thus clearly different from that of proteins encoded by the ~290 cloned plant resistance genes, which were largely (73%) extracellular or intracellular immune receptors (Supplementary Table 1). To explore the unusual structure of Sr43 in more detail, we used the AlphaFold artificial intelligence–augmented system to generate a three-dimensional (3D) model[26] (Supplementary Data 1). We determined that Sr43 adopts a modular structure, with the kinase and the two DUFs separated by flexible linker loops (Fig. 1e). The kinase domain contained α-helices and antiparallel β-strands, whereas the DUFs were entirely α-helical. We compared the predicted structure of the Sr43 protein to those in the Protein Data Bank[27]. This identified structural similarities between DUF668 and some receptor-like protein kinase–like proteins outside of their kinase domains. We searched for ATP-binding sites using the small molecule docking program HADDOCK[28] and identified one high-confidence ATP-binding site in DUF668 (Fig. 1f, Supplementary Table 11 and Supplementary Data 2).

We cloned a 14 kb genomic *Sr43* fragment including 3.2 kb of upstream and 2.5 kb of downstream regulatory sequence (Fig. 2a) (Supplementary Table 12) and introduced the resulting binary construct into the wheat cultivar Fielder. We obtained one primary (T_0) transgenic plant and on the basis of quantitative PCR (qPCR) identified a genetically stable line with an estimated two copies of *Sr43* (Supplementary Tables 13–14 and Supplementary Fig. 17). We tested homozygous T_1 and T_2 lines against a geographically and phenotypically diverse panel of 11 *Pgt* isolates from North America, the Middle East, Europe and Africa. In ten cases, the *Sr43* transgenic and wild-type introgression lines were resistant, whereas the cultivars Chinese Spring (the introgression parent) and Fielder were susceptible (Fig. 2b, Extended Data Fig. 4a,b and Supplementary Table 15). By contrast, the *Pgt* isolate 75ND717C was intermediately virulent on the *Sr43* introgression and transgenic lines (Fig. 2b). For *Pgt* isolate 69MN399, we compared the phenotype at 21 °C and 26 °C and noticed a marked reduction in *Sr43*-mediated resistance at the higher temperature, in line with previous observations[29] (Fig. 2c). Taken together, these results confirm (1) the wide-spectrum efficacy of *Sr43* (ref. 29), (2) that a 14 kb *Sr43* genomic fragment is sufficient for function and (3) that the transgenic line faithfully recapitulates the race-specific and temperature-sensitive resistance of wild-type *Sr43*.

We searched for Sr43 homologs to investigate its evolutionary origin. We identified proteins harboring either the kinase domain or the two DUFs alone across the Poaceae family spanning 60 million years of evolution (Supplementary Tables 16 and 17 and Extended Data Figs. 5 and 6). We detected the Sr43 protein domain arrangement only within the *Thinopyrum*, *Triticum*, *Aegilops* and *Secale* genera of the Triticeae tribe but not within *Hordeum*, suggesting that *Sr43* probably arose between 6.7 and 11.6 million years ago (Fig. 3 and Supplementary Table 18). In those lineages lacking a clear *Sr43* homolog, we mapped genes encoding the kinase and DUFs present in Sr43 to different

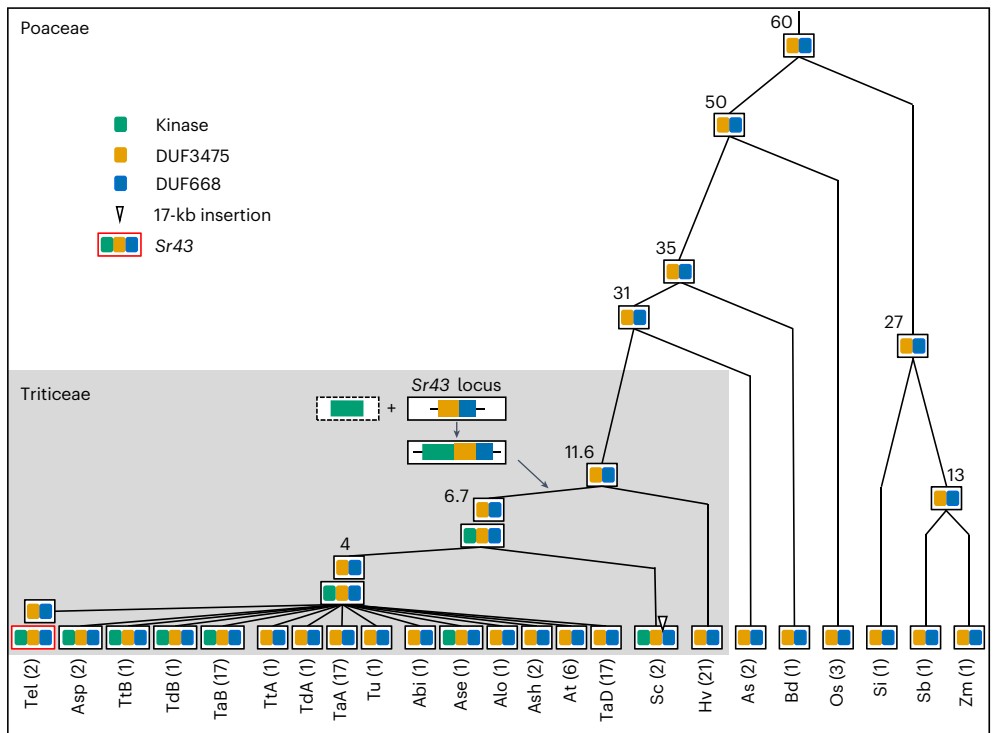

**Fig. 3 | Phylogenetic tree of the Poaceae showing the origin and distribution of *Sr43* orthologs.** The age of the last common ancestor in millions of years is indicated at each node on the basis of ref. 23. The Triticeae clade is highlighted in gray. Species are indicated at the bottom and abbreviated as follows: Tel, *Thinopyrum elongatum*; Asp, *Ae. speltoides*; TtB, *Triticum turgidum* ssp. *durum* B genome; TdB, *T. dicoccoides* B genome; TaB, *T. aestivum* B genome; TtA,

*T. turgidum* ssp. *durum* A genome; TdA, *T. dicoccoides* A genome; TaA, *T. aestivum* A genome; Tu, *T. urartu*; Abi, *Aegilops bicornis*; Ase, *Ae. searsii*; Alo, *Ae. longissima*; Ash, *Aegilops sharonensis*; At, *Ae. tauschii*; TaD, *T. aestivum* D genome; Sc, *Secale cereale*; Hv, *Hordeum vulgare*; As, *Avena sativa*; Bd, *Brachypodium distachyon*; Os, *Oryza sativa*; Si, *Setaria italica*; Sb, *Sorghum bicolor*; Zm, *Zea mays*. The number of genomes analyzed for each species is indicated in parentheses.

chromosomes (for example, *Sorghum bicolor*, *Zea mays*, *T. urartu* and *Ae. sharonensis*) or on the same chromosome but 6–36 megabases (Mb) apart (*Ae. tauschii* and *Setaria italica*), suggesting that the recruitment of the kinase domain to the DUFs at the *Sr43* locus involved an ectopic recombination event (Supplementary Table 18). In *Thinopyrum elongatum*, the ancestral state and *Sr43* were retained as an intraspecies polymorphism; some species of *Aegilops* and *Triticum* retained the ancestral state (for example, *Ae. tauschii*), whereas others retained the *Sr43* innovation (for example, the *T. aestivum* and *T. durum* B genomes) (Fig. 3).

In summary, we cloned the wheat stem rust resistance gene *Sr43*, which encodes a protein kinase fused to two DUFs. Of the 82 Triticeae resistance genes cloned to date, most encode NLRs (*n* = 46), followed by protein kinase fusion proteins (*n* = 15) (Supplementary Table 19). Of the latter, seven are tandem kinases, whereas Sr43, Pm4, Snn3, Sm1, Tsn1, Yr36, Rpg5 and Lr9 are single or tandem protein kinases fused to different domains[16–22] (Extended Data Fig. 7 and Supplementary Table 20). Little is known about the function of kinase fusion proteins but most confer race-specific resistance that is phenotypically indistinguishable from NLR-mediated resistance. Their encoding genes do not fall into the *Lr34/Lr67* category of adult, broad-spectrum and multipathogen resistance[30,31]. To explain the role of these kinases in resistance, we sought clues from NLRs whose modus operandi is now well understood. NLRs can act as guards that monitor host components targeted by pathogen effectors[32]. These guards detect the interaction between an effector and its target, leading to a conformational change in the NLR that triggers downstream defense responses. This tripartite interaction creates an evolutionary 'tug-of-war' that imposes selective pressure (1) on the effector to evade detection by the NLR while maintaining its ability to coerce the pathogenicity target, (2) on the NLR to recognize new effector variants and (3) on the pathogenicity target

to avoid being disrupted by the effector while maintaining its cellular function. Duplication of the pathogenicity target can release it from this functional constraint and provide a 'decoy' for the effector. This diversification may also result in the decoy behaving genetically as the resistance gene[33]. In about 10% of all NLRs, the decoy has become integrated into the NLR itself[34]. Such a guardee–decoy fusion ensures that both components are inherited as a single operational unit.

By extrapolation, protein kinase fusion proteins may be pathogenicity targets that are guarded by NLRs. All protein kinase fusion proteins have one apparent functional kinase that is fused to a second, typically non-functional, kinase domain but sometimes to an altogether different domain, as in for example Sr43, Lr9 and Pm4 (Extended Data Fig. 7). Perhaps as with those NLRs that carry an integrated decoy, this second domain might be an integrated decoy, while the apparent functional kinase exerts the signaling function. Indeed, plants produce various enzymes, including protein kinases with different integrated domains, to catalyze reactions of various substrates. In the case of protein kinase resistance proteins, the integrated domain would define the specific substrates of pathogen Avr proteins, whereas the kinase would catalyze the phosphorylation of the Avr protein, the integrated domain, itself, or a third signaling partner to trigger downstream defense, possibly via an NLR guard (Extended Data Fig. 8a). EMS mutagenesis of *Yr15*, *Pm24*, *Sr62* and *Sr43* has shown a preponderance of missense mutations affecting the kinase active site or ATP-binding pocket of the apparent functional kinase domain (Extended Data Fig. 7), supporting the notion that kinase-mediated signaling is required for function. Alternatively, Sr43 (and by extrapolation other kinase fusion resistance proteins)[35,36] may function without an NLR cosignaling partner (Extended Data Fig. 8b).

The transgenic expression of *Sr43* in a different background allowed us to confirm the broad-spectrum efficacy of *Sr43*, highlighting

its potential value in resistance breeding. However, it is possible to obtain gain-of-virulence pathogen mutants that have lost *AvrSr43* function under laboratory conditions[37]. Therefore, *Sr43* should be used in combination with other broad-spectrum resistance genes to maximize its longevity in the field.

## Online content

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

## Methods

### Mutant collection development

We mutagenized 2,700 seeds of the wheat–*Th. elongatum* introgression line RWG34 containing *Sr43* (ref. 29). Dry seeds were incubated for 16 h with 200 ml of a 0.8% (w/v) EMS solution with constant shaking on a Roller Mixer (Model SRT1, Stuart Scientific) to ensure maximum homogenous exposure of the seeds to EMS. The excess solution was then removed and the seeds were washed three times with 400 ml of tap water. The $M_1$ seeds were grown in the greenhouse and the seeds of $M_2$ families (single heads) were collected. Eight seeds per $M_2$ family were phenotyped with the *Pgt* isolate, race TPMKC. The $M_3$ seeds derived from susceptible $M_2$ plants were also tested to confirm that the $M_2$ susceptible plants were true non-segregating mutants. To rule out seed contamination, ten mutants were verified using genotyping-by-sequencing (GBS)[38]. GBS data from the background (Chinese Spring), donor species (*Th. elongatum*, accession PI531737 from USDA-ARS GRIN), the Chinese Spring–*Th. elongatum Sr43* introgression line (RWG34) and the mutant lines were mapped to the reference genome sequence of Chinese Spring[39] using BWA mem (v.0.7.12) with standard parameters[40]. Mapping results were sorted and converted to mpileup format using SAMtools[41] (v.0.1.19). The mpileup files were examined with a previously published custom script linked to Zenodo[42] to calculate the percentage of single-nucleotide polymorphisms from the donor that were shared with the introgression line per given interval. Several interval lengths were tested; a clear signal was observed for 10 Mb intervals.

### Chromosome flow sorting

Suspensions of mitotic metaphase chromosomes were prepared from root tips of the *Sr43* introgression line and eight EMS mutants as described by ref. 43 and ref. 44. Briefly, root tip meristem cells were synchronized using hydroxyurea, accumulated in metaphase using amiprophos-methyl and fixed in 2% (v/v) formaldehyde at 5 °C for 20 min. Intact chromosomes were released by mechanical homogenization of 100 root tips in 600 µl of ice-cold LB01 buffer[45]. GAA microsatellites were labeled on isolated chromosomes by fluorescence in situ hybridization in suspension (FISHIS) using 5′-FITC-GAA$_7$-FITC-3′ oligonucleotides (Sigma) according to ref. 46 and chromosomal DNA was stained with 2 µg ml$^{-1}$ of 4′,6-diamidino 2-phenylindole (DAPI). Chromosome analysis and sorting were conducted using a FACSAria II SORP flow cytometer and sorter (Becton Dickinson Immunocytometry Systems). Bivariate flow karyotypes plotting FITC and DAPI fluorescence were acquired for each sample and chromosomes were sorted at rates of 20–40 particles per second. Two batches of 55,000 and 70,000 copies of chromosome 7D with the *Th. elongatum* chromosome segment carrying *Sr43* were sorted from the wild-type *Sr43*-WT and one batch of 14,000–66,000 copies was sorted from the eight mutants into PCR tubes containing 40 µl of sterile deionized water (Supplementary Table 3).

Chromosome content of flow-sorted fractions was estimated by microscopy observations of 1,500–2,000 chromosomes sorted into a 10 µl drop of PRINS buffer containing 2.5% (w/v) sucrose[47] on a microscope slide. Air-dried chromosomes were labeled by FISH with probes for the pSc119.2 repeat, Afa family repeat and 45S ribosomal DNA that allowed identification of all wheat and *Th. elongatum* chromosomes according to ref. 48. To determine chromosome contents and purity in the sorted fractions, at least 100 chromosomes in each flow-sorted sample were classified following the karyotype described by ref. 49.

### Sequencing and assembly of 7D–7E translocation chromosomes

The two separately flow-sorted chromosomal samples of the wild-type genotypes were used for preparation of two sequencing libraries. Chromosomal samples were treated with proteinase K (60 ng µl$^{-1}$), after which DNA was purified without amplification. Chromosomal samples flow sorted from the mutants were treated similarly but their DNA contents were amplified to 2.5–12.6 µg by multiple displacement amplification (Supplementary Table 3) using an Illustra GenomiPhi v.2 DNA Amplification Kit (GE Healthcare) as described by ref. 50 and sequenced by Novogene. For the *Sr43* wild-type genotypes, 20 ng of non-amplified DNA was fragmented in a 20 µl volume using a Bioruptor Plus (Diagenode) five times for 30 s on the HIGH setting. Libraries for sequencing were prepared from fragmented DNA using an NEBNext Ultra II DNA Library Prep Kit for Illumina with the following modifications: (1) size selection was directed for larger final library size (~1,000 bp) and (2) PCR enrichment was done with six PCR cycles. Libraries were sequenced on a HiSeq2500 platform using a HiSeq Rapid SBS Kit v.2 as 250 bp paired-end reads. The raw data were trimmed for low-quality bases using Trimmomatic[51] and assembled into scaffolds with Meraculous[52] (v.2.0.5) using 111 nucleotide *k*-mers. Scaffolds shorter than 1 kb were eliminated. The assembly contained 168,523 scaffolds with a total assembly length of 1.29 gigabase (Gb). Among them, 25,581 scaffolds were longer than 13.9 kb with a total length 631.8 Mb.

### Candidate gene identification

Eight susceptible mutants derived from independent $M_2$ families were selected for MutChromSeq mapping[53]. The raw reads from the eight mutants were individually mapped to the 10 kb chopped scaffold fragments using BWA[40] (v.0.7.12) and SAMtools[41] (v.1.8). One fragment was identified as having a single nucleotide mutation in all mutants. We calculated the probability of this being the candidate gene using formula number 4 developed by ref. 12, with 2,598 bp of *Sr43* coding sequence (CDS), assuming the average gene CDS is 1,000 bp in length and that chromosome 7D has 5,822 genes. All identified mutations were G-to-A or C-to-T transition mutations, which are typical of EMS mutagenesis.

### RNA extraction and *Sr43* annotation

Total RNA was extracted from the Chinese Spring–*Th. elongatum Sr43* introgression line with an RNeasy Plant Mini Kit (catalog no./ID 74904, Qiagen) following the manufacturer's protocol and digested with Dnase I (Roche). RNA-seq was performed by Novogene. The RNA-seq reads were trimmed with Trimmomatic (http://www.usadellab.org/cms/?page=trimmomatic). Hisat2 (v.2.1.0)[54] was used to map the short reads onto the *Sr43* genomic sequence. The SAM output file was converted into a BAM file using SAMtools[41] (v.1.8) (http://www.htslib.org/) and sorted according to their position along the *Sr43* genomic sequence and indexed for visualization by IGV (https://software.broadinstitute.org/software/igv/). To determine the alternative splicing of *Sr43*, we constructed a full-length cDNA library using a SMARTer PCR cDNA Synthesis kit (catalog no. 634926, Clontech/TaKaRa). Transcripts corresponding to each of the four splice variants were identified by Sanger sequencing of 20 clones obtained from transformation of long-range PCR on the full-length cDNA library with primers specific to the *Sr43* 5′ and 3′ ends (Supplementary Table 5).

### Engineering of the *Sr43* binary vector construct

On the basis of the gene annotation, three overlapping segments of the *Sr43* gene were PCR-amplified (Supplementary Table 12) with high-fidelity Q5 DNA polymerase (NEB) following the manufacturer's instructions. The PCR products were purified with a QIAquick PCR Purification kit (QIAGEN) and A-tailed using *Taq* DNA polymerase before being cloned into the pCR2.1 vector (TOPO PCR Cloning Kits-K202020, Thermo Fisher Scientific). The positive clones were digested with three sets of restriction enzymes, NotI, NotI-PvuI and PvuI-PmeI (NEB), to generate *Sr43* fragment parts 1, 2 and 3, respectively. The digested fragments were gel-purified and then parts 2 and 3 were combined in a three-way ligation reaction with the binary vector pGGG-AH-NotI/PmeI[12] digested with NotI and PmeI, using T4 DNA ligase (M0202S, NEB). Subsequently, the binary construct was linearized with NotI and part 1 was dropped in. A positive clone with part 1 in the correct

orientation, pGGG-*Sr43*, was verified by Sanger sequencing. The pGGG-*Sr43* is available from Addgene under accession number 186974.

## Wheat transformation

The binary construct pGGG-*Sr43* was transformed into wheat cv. Fielder using *Agrobacterium tumefaciens*-mediated transformation[55]. The *Sr43* copy number was postulated by testing the copy number of the *hygromycin B phosphotransferase* selectable marker in $T_0$ and $T_1$ plants by iDNA Genetics using qPCR[56]. From a non-segregating genetically stable $T_1$ family we advanced a $T_2$ line (BW_30183) for further copy number testing. We designed gene-specific primers for *Sr43*, the *hygromycin B phosphotransferase* selectable marker gene and single-copy, three-copy and six-copy wheat endogenous control genes (Supplementary Table 13). The primer sequences for the endogenous genes were designed on the basis of the cv. Fielder reference genome[57]. DNA was extracted from a single $T_3$ plant (derived from the $T_2$ family BW_30183) using the Qiagen genomic DNA extraction kit (Qiagen, catalog no. 19060) with 500 per g columns (Qiagen, lot 169047970) following the QIAGEN Genomic DNA Handbook. The qPCR was done in a 10 µl reaction with 1X SsoAdvanced Universal SYBR Green Supermix (BioRad), 0.5 µM primer and 2 ng µl$^{-1}$ of DNA using an initial denaturation at 95 °C for 3 min, followed by a denaturation at 95 °C for 15 s and annealing + extension at 60 °C for 30 s, for 40 cycles on a CFX96 Real-Time PCR system. The *Sr43* gene copy number was calculated on the basis of the endogenous reference genes (Supplementary Fig. 17 and Supplementary Table 14).

## Stem rust phenotyping

The stem rust tests were carried out in a greenhouse or in growth chambers. The greenhouse/growth chambers were maintained at 21 °C with a 14 h light period and ~40% relative humidity. Plants were inoculated with *P. graminis* f. sp. *tritici* when the second leaf was fully expanded, 10–12 days after sowing, at a rate of ~0.12 mg of spores per plant. After a 16 h incubation period in the dark under high humidity (100%) conditions, inoculated seedlings were returned to the greenhouse/growth chamber and then scored for reaction to stem rust 12–14 days later. The infection types were recorded using the Stakman scale[58]. For temperature sensitivity tests, the high temperature was set to 26 °C. The *Pgt* races used in this study were TPMKC (isolate 74MN1409) from the United States; QTHJC (isolates 75ND717C and 69MN399) from the United States; TKTTF (isolate ET11a-18) from Ethiopia; TTKTT (isolate KE184a/18) from Kenya; TKTSC (isolate IS no. 2079), TTTTF (isolate IS no. 2127) and TTTTC (isolate IS no. 2135) from Israel; TKTTF (isolate FR68-20) from France; TTRTF (isolate IT16a-18) from Italy; TKTTF (isolate UK-01) from the United Kingdom; and TRTTF (isolate 14GEO189-1) from Georgia (Supplementary Table 15).

## Protein homology searches

We used InterPro v.88.0 to search for protein family domains in Sr43, for example, a transmembrane domain[59]. To check for the presence of myristoylation sites and nuclear localization signals, we used Myristoylator[60] and cNLS mapper[61], respectively (accessed 11 March 2023).

## Sr43 protein 3D modeling and ATP-binding site prediction

We used the open source code of AlphaFold v.2.0 (ref. 26) and the supercomputer of King Abdullah University of Science and Technology, Shaheen II (https://www.hpc.kaust.edu.sa/) through the multi-node system Ibex (https://www.hpc.kaust.edu.sa/ibex). We input the amino acid sequence of Sr43 and the output was five unrelaxed, five relaxed and five ranked models in .pdb format. We used the ranked_1.pdb model that contains the predictions with the highest confidence with the best local distance difference test (lDDT) score standing at 70.76. We next input the ranked_1.pdb model obtained from Alphafold and each domain separately into the protein structure comparison server Dali[27].

We used HADDOCK2.4, a web server for small molecule binding site screening[28], to screen the DUF2 domain for potential ATP-binding sites. The input files consisted of the DUF2 domain of Sr43 after removing all loops from the .pdb file and ATP in .pdb format. The settings used were:

Define randomly ambiguous interaction restraints from accessible residues−ON
Number of structures for rigid body docking−10,000
Number of structures for semiflexible refinement−400
Number of structures for the final refinement−400
Clustering method (RMSD or fraction of common contacts (FCC))−RMSD
RMSD cutoff for clustering (recommended: 7.5 A for RMSD, 0.60 for FCC)−2.0
Evdw 1−1.0
Eelec 3−0.1
Initial temperature for second TAD cooling step with flexible side-chain at the interface−500
Initial temperature for third TAD cooling step with fully flexible interface−300
Number of MD steps for rigid body high temperature TAD−0
Number of MD steps during first rigid body cooling stage−0

The output files were ten clusters of different predicted ATP-binding sites. The cluster with the best prediction score (*Z*-score) was cluster 6.

## Expression and purification of recombinant Sr43 protein

The native CDS of *Sr43* plus two additional nucleotides (CC) at the beginning of the CDS (to maintain the open reading frame with His$_6$ tag) was commercially synthesized (Twist Bioscience) and cloned into the Gateway entry vector pTwist_ENTR. For recombinant protein expression, *Sr43* was transferred into the expression vector pDEST-His$_6$-MBP by Gateway LR clonase reaction (Invitrogen). The resulting clone was verified by Sanger sequencing.

His$_6$-MBP-Sr43 tagged protein was expressed in the *E. coli* Rosetta strain by growing the bacterial culture to an optical density OD$_{600}$ of 0.8 at 37 °C and then inducing the protein expression by addition of 0.5 mM isopropyl β-D-thiogalactopyranoside at 18 °C and further incubating the culture for 14–16 h. The recombinant protein was purified under native conditions using Ni-NTA agarose beads (Invitrogen catalog no. R901-15) following the manufacturer's instructions.

## In vitro kinase reaction and phosphosite identification

The buffer composition of the purified His$_6$-MBP-Sr43 protein was changed to the kinase reaction buffer (20 mM Tris-HCl pH 7.5, 10 mM MgCl$_2$, 5 mM EGTA, 1 mM DTT and 50 µM ATP) using PD10 desalting columns (GE Healthcare). His$_6$-MBP-Sr43 was mixed with a commercial substrate maltose-binding protein DNA gyrase (Prospec Protein Specialists PRO-616) and incubated at ambient temperature for 30 min. After adding SDS-sample buffer to stop the reaction, the protein was denatured by boiling at 95 °C for 10 min. SDS–polyacrylamide gel electrophoresis was used to resolve protein samples. The gel was stained with SimplyBlue SafeStain (Novex cat. no. LC6065) and the band that corresponded to the protein of interest was excised, cut into pieces of 0.5 mm$^3$ and destained with four sequential washes of 15 min each with acetonitrile and 100 mM NH$_4$HCO$_3$. The proteins in the gel pieces were reduced with 10 mM Tris (2-carboxyethyl) phosphine hydrochloride (TCEP, C-4706 Sigma) in 100 mM NH$_4$HCO$_3$ at 37 °C for 1 h. Then the reduced disulfide bonds were alkylated with 50 mM iodoacetamide at ambient temperature for 30 min. Following reduction and alkylation of proteins, they were digested with trypsin (porcine trypsin, Promega) at 37 °C overnight. Formic acid was added to a final concentration of 1% to stop the digestion and the tryptic peptides were recovered by incubating the gel pieces in acetonitrile. The recovered peptides were desalted using Sep-Pak C18 1 ml vac cartridge (Waters SKU: WAT023590) and analyzed

by liquid chromatography with tandem mass spectrometry (LC-MS/MS) (Supplementary Fig. 14). Peptide samples were separated using a C18 column linked to an Orbitrap Fusion Lumos mass spectrometer (Acclaim PepMap C18, 25 cm length 75 m I.D. 3 m particle size, 100 porosity, Dionex). The LC gradient increased from 5% solvent B (water/ACN/formic acid, 20/80/0.1, v/v/v) to 45% solvent B over 45 min, then to 90% solvent B for 10 min. Using HCD fragmentation in the Orbitrap Fusion Lumos instrument, the MS instrument recorded fragmentation spectra on the top ten peptides. Using the msConvert interface, the RAW data files were converted to MGF files. The Mascot server was used to conduct database searches and the following criteria were used: (1) database containing the amino acid sequences of Sr43, MBP and contaminant proteins; (2) enzymatic specificity (trypsin permitting two allowed missed cleavages); (3) cysteine residues are fixedly modified (carbamidomethyl); (4) phosphorylation of S, T and Y residues may be variably modified; (5) precursor masses are tolerable to 5 ppm; (5) fragment ions are tolerable to 0.02 Da. Mascot and MD scores were used to filter the findings. The peptides identified for determining the protein coverage of maltose-binding protein are shown when incubated alone and when incubated with His6-MBP-Sr43 (Supplementary Figs. 15 and 16).

### Sr43 protein localization in *N. benthamiana*

To generate the 35S:*Sr43-GFP* construct, a codon-optimized open reading frame of *Sr43* splice version 1 was synthesized and ligated into pDONR221 (Twist Bioscience). The entry clone was then introduced into the binary vector pB7FGW2,0 by single Gateway LR reaction (Invitrogen). The 35S:*PLSP2A-mRFP* construct was generated by combining an entry clone of the *PLSP2A* CDS with the *p35S* promoter and the *mRFP* fluorescent reporter using sequential Gateway cloning.

The 35S:*Sr43-GFP* and 35S:*PLSP2A-mRFP* constructs were transferred into *A. tumefaciens* strain GV3101 and infiltrated into tobacco leaves as described in ref. 62. The GFP signal was excited at 488 nm and detected between 500 and 535 nm. The mRFP signal was excited at 555 nm and detected between 566 and 646 nm. Images were acquired using an inverted Leica SP8 Stellaris FALCON with an HC PL APO 63× 1,2 W CORR UVIS CS2 objective.

### Sr43 protein localization in wheat protoplasts

To generate the pZmUbi:*GFP-Sr43* construct, the splice version 1 *Sr43* CDS was subcloned from the pDONR221 mentioned above into the pJET1.2 vector (Thermo Fisher) as a level I module for further Golden Gate cloning[63]. The level II expression construct was assembled with the level II backbone (BB10), the level I ZmUbiquitin promoter, the level I *GFP*-tag, the level I *Sr43* and the level I NOS terminator via a BsaI cut-ligation reaction[63]. The pZmUbi:*NLS-mCherry* construct was generated in the same way by combining the level II backbone, the level I ZmUbqiuitin promoter, the level I nuclear localization sequence, the level I *mCherry* and the level I NOS terminator. The pZmUbi:*GFP* control was generated by transferring the GFP CDS via LR clonase reaction to pZmUbi:GW[64].

Plasmid DNAs were purified from *E. coli* harboring pZmUbi: *GFP-Sr43*, pZmUbi:*NLS-mCherry*, pZmUbi:*GFP* with NucleoBond Xtra Maxi Plus Kit (Macherey-Nagel). Mesophyll cells were isolated from 9-day-old wheat seedlings (cv. Fielder) grown in short-day conditions (8 h light, 16 h dark). Protoplast isolation and transfection were performed as described in ref. 64.

Florescence was observed with a Carl Zeiss LSM upright 880 Axio Imager 2 confocal microscope with a Plan-Apochromat 63×/1.4 Oil DIC M27 objective. GFP was excited using an argon laser (488 nm) and detected between 494 and 552 nm. The mCherry was excited using a Diode Pumped Solid State laser (561 nm) and florescence was detected between 596 and 649 nm.

### Phylogenetic analysis

We constructed a phylogenetic tree on the basis of the aligned protein sequences of 100 best hits (ID ≥75%) of the kinase domain sequence and Sr43 DUF region sequence against the NCBI protein database. The phylogenetic tree (neighbor-joining method) for kinase and DUF domains were computed with Clustal Omega (https://www.ebi.ac.uk/Tools/msa/clustalo/) and drawn with iTOL (https://itol.embl.de/).

### Reporting summary

Further information on research design is available in the Nature Portfolio Reporting Summary linked to this article.

### Data availability

The datasets generated during and/or analyzed in the current study are publicly available as follows. The sequence reads were deposited in the European Nucleotide Archive under project numbers PRJEB52878 (GBS data), PRJEB51958 (chromosome flow-sorted data) and PRJEB52088 (RNA-seq data). The *Sr43* gene and transcript sequence were deposited in NCBI Genbank under accession number ON237711. The *Sr43* chromosome assembly has been deposited in Zenodo (https://doi.org/10.5281/zenodo.6777941). The following public databases/datasets were used in the study: Chinese Spring reference genome[39], Gramene (http://www.gramene.org/), https://ensembl.gramene.org/Multi/Tools/Blast, https://wheat.pw.usda.gov/GG3/blast, BLAST non-redundant protein sequence (https://blast.ncbi.nlm.nih.gov/Blast.cgi?PROGRAM=blastx&PAGE_TYPE=BlastSearch&LINK_LOC=blasthome), Taxonomy Browser (https://www.ncbi.nlm.nih.gov/Taxonomy/Browser/wwwtax.cgi?id=1437183), AlphaFold[26] (https://alphafold.ebi.ac.uk), Dali[27] (http://ekhidna2.biocenter.helsinki.fi/dali/) and HADDOCK[28] (https://www.bonvinlab.org/education/HADDOCK-binding-sites/.

### Code availability

The scripts used in these analyses have been published in GitHub (https://github.com/steuernb/GBS_introgression_line_analysis) and linked with Zenodo (https://zenodo.org/badge/latestdoi/394326594)[42].

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

## Acknowledgements

We thank Y. Wang for help with phenotyping and compiling Supplementary Table 19; E.S. Vande Loo for media preparation; H. Zhang and A.W. Weatherhead for help with mass spectrometry (all KAUST, Saudi Arabia); Y. Jin (USDA-ARS, Minnesota, USA) for use of *Pgt* isolates 74MN1409, 75ND717C, 69MN399 and 14GEO189-1; M. van Slageren (Kew, UK) for help with species nomenclature; S. Saile and L. Rohr (University of Tübingen, Germany) for pZmUbi and NLS Golden Gate modules; Z. Dubská, R. Šperková and J. Weiserová for preparation of chromosome samples for flow cytometry; and M. Said and P. Cápal for chromosome sorting (all IEB, Czech Republic). This research was supported by the NBI Research Computing group and the Informatics Platform at the John Innes Centre, UK, and financed by funding from the 2Blades Foundation, USA, to B.J.S. and B.B.H.W.; the Biotechnology and Biological Sciences Research Council (BBSRC) Designing Future Wheat Cross-Institute Strategic Programme to B.B.H.W. (BBS/E/J/000PR9780); Marie Curie Fellowship grant award 'AEGILWHEAT' (H2020-MSCA-IF-2016-746253) and the Hungarian National Research, Development and Innovation Office (K135057) to I.M.; ERDF project 'Plants as a tool for sustainable global development' (no. CZ.02.1.01/0.0/0.0/16_019/0000827) to J.B., K.H., M.K. and J.D.; King Abdullah University of Science and Technology to B.B.H.W., Ł.J., I.B. and H.H.; the Lieberman-Okinow Endowment at the University of Minnesota to B.J.S.; the Daimler and Benz Foundation, by the German Research Foundation (DFG) CEPLAS (EXC 2048/1—Project-ID: 390686111) and the DFG Emmy Noether Programme (SA 4093/1-1) to I.M.L.S.; the Gordon and Betty Moore Foundation through grant GBMF4725 to the 2Blades Foundation and J.D.G.J.; and the Gatsby Charitable Foundation to J.D.G.J.

## Author contributions

G.Y. generated the mutant population. R.J., B.J.S. and N.K. screened and verified mutants. B.S., S.W. and J.P. genotyped mutants. I.M., M.K. and J.D. performed chromosome flow sorting and amplification. G.Y., K.H. and J.B. did the chromosome assembly. G.Y. performed bioinformatics analyses to identify *Sr43*. G.Y. extracted RNA and determined *Sr43* gene structure and alternative splicing. G.Y., S.G. and Ł.J. annotated Sr43. S.G. and Ł.J. performed 3D modeling analyses. M.S. developed the binary vector. G.Y. engineered the *Sr43* construct. S.H. and W.H. transformed *Sr43* into wheat. G.Y., O.M., M.P. and M.N.R. phenotyped the transgenic lines. F.R.A. and I.B. (in *N. benthamiana*) and Y.L.W., E.E.C., T.N. and I.M.-L.S. (in wheat) determined Sr43 localization. N.R. and H.H. performed kinase assay. G.Y. performed phylogenetics and synteny. O.M., S.X., M.N.R., M.R., A.S., J.D.G.J., C.G., Y.Y. and T.L.R. provided germplasm, rust cultures or scientific support and advice. B.B.H.W., G.Y. and B.J.S. conceived the study. G.Y. and B.B.H.W. drafted the manuscript. All coauthors read and approved the final manuscript. Authors are grouped by institution in the author list, except for the first six and last six authors.

## Competing interests

G.Y. and B.B.H.W. are inventors on the US provisional patent application 63/250,413 filed by 2Blades and relating to the use of *Sr43* for stem rust resistance in transgenic wheat. T.L.R. was employed by the 2Blades Foundation, which cofunded the work. The remaining authors declare no competing interests.

## Additional information

**Extended data** is available for this paper at https://doi.org/10.1038/s41588-023-01402-1.

**Correspondence and requests for materials** should be addressed to Brian J. Steffenson or Brande B. H. Wulff.

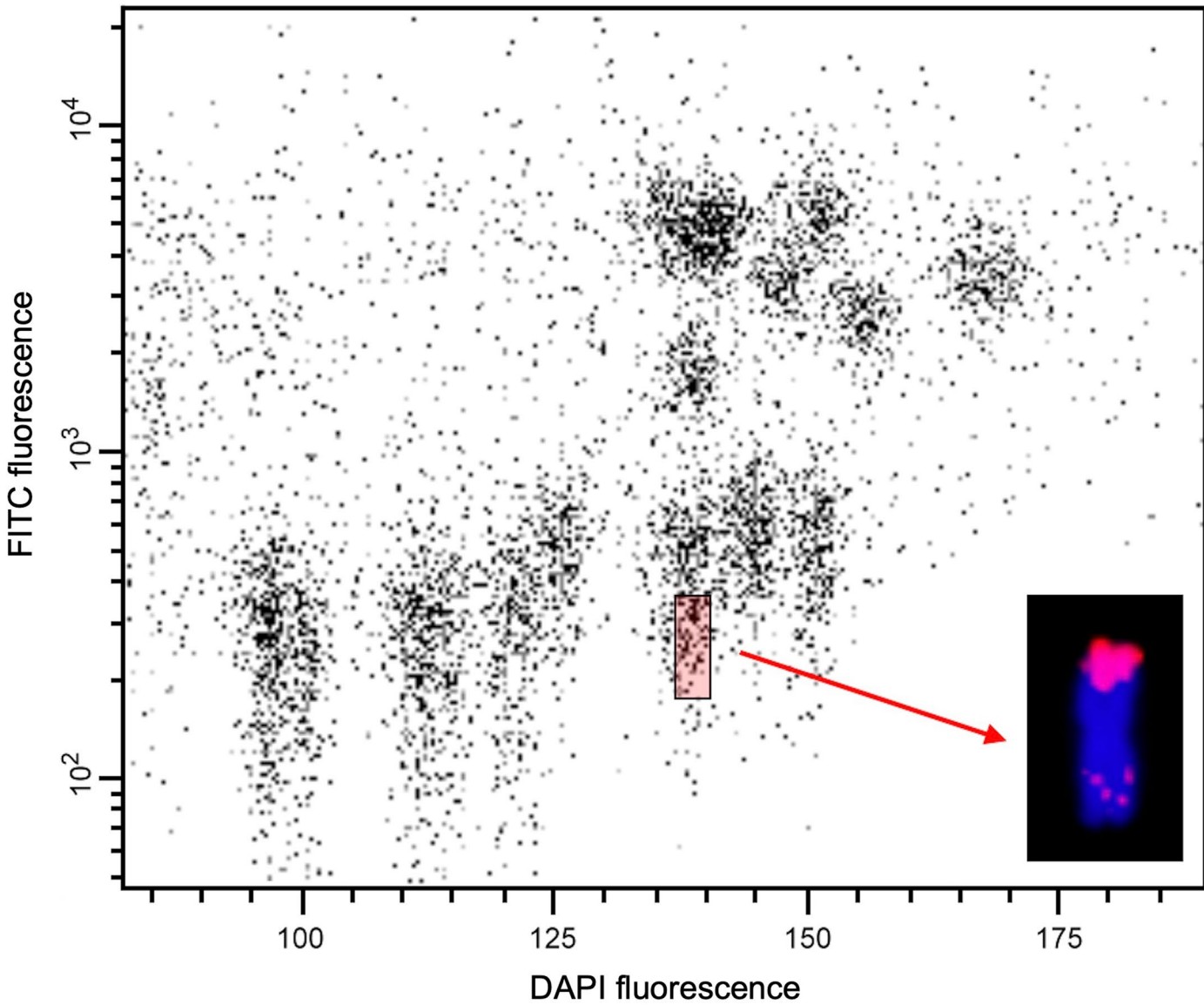

**Extended Data Fig. 1 | Bivariate flow karyotype of mitotic metaphase chromosomes isolated from the wheat-*Th. elongatum* 7D/7el2 translocation line carrying the *Sr43* resistance gene.** DAPI (x-axis) vs. FITC (y-axis) dot plot was obtained after the analysis of DAPI-stained chromosome suspensions labeled by FISHIS with FITC-conjugated probes for GAA and ACG microsatellites. The 7D/7el2 translocation chromosomes were sorted from the sorting window shown as red rectangle at purities of 60–65%. Inset: 7D/7el2 translocation chromosome after FISH with probes for pSc119.2 repeat (green), Afa family repeat (red) and 45S rDNA (yellow) that was used to identify chromosomes in the sorted fraction. Chromosomes were counterstained by DAPI (blue).

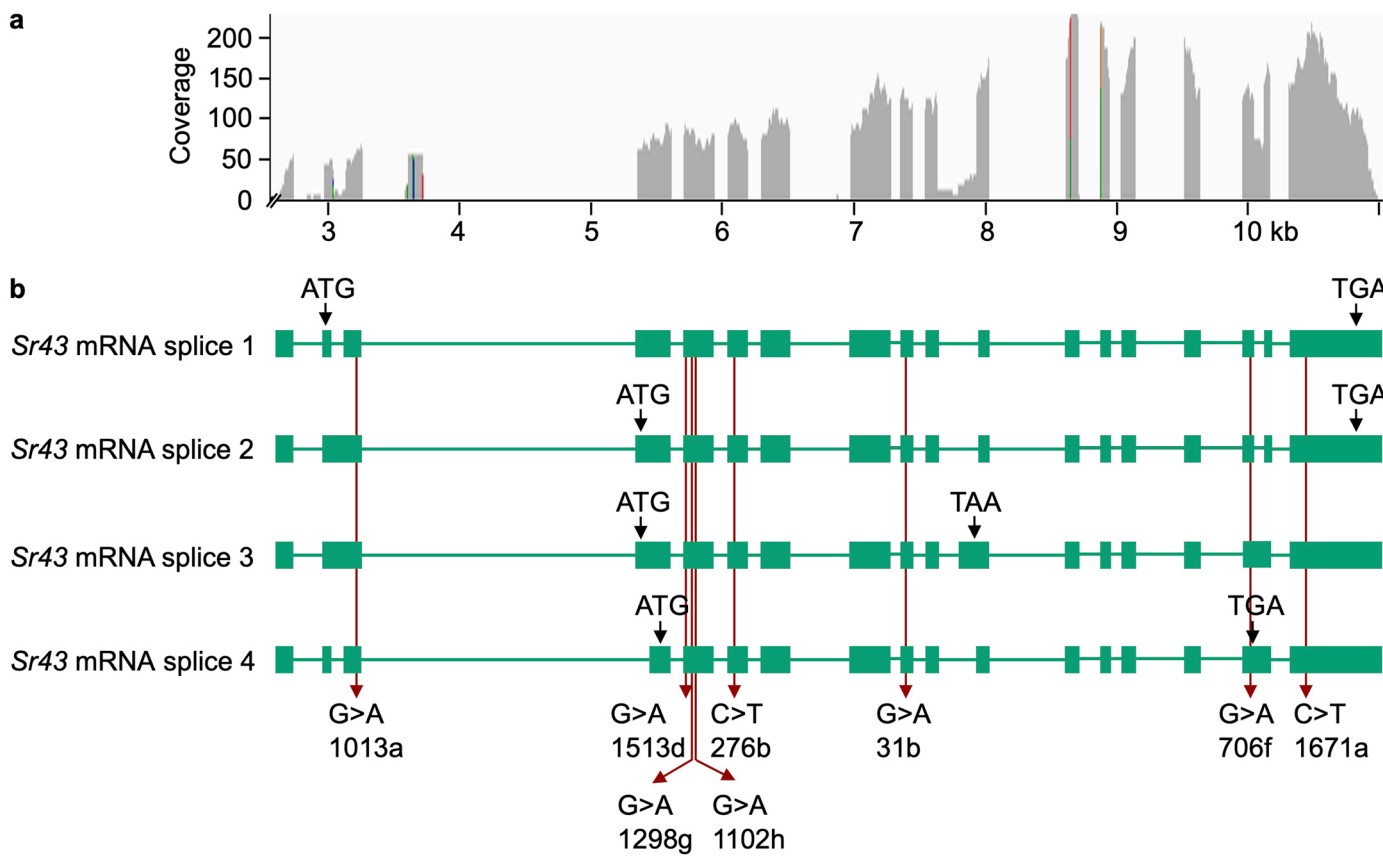

**Extended Data Fig. 2 | *Sr43* transcript and alternative splicing. a**, Mapping of RNA-seq reads from the leaf transcriptome onto the *Sr43* locus. **b**, mRNA splice variants identified by sequencing full-length cDNA clones. Introns and exons are represented by lines and boxes, respectively. Splice variants 1, 2, 3 and 4 have 8, 7, 5 and 6 EMS-induced mutations in the coding sequence, respectively.

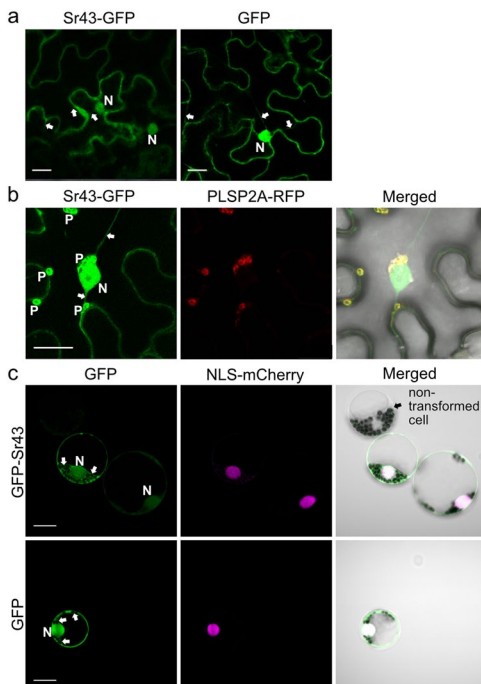

**Extended Data Fig. 3 | Sr43 localizes to the nucleus and plastids. a**, *Agrobacterium*-mediated transient expression of a *Sr43-GFP* fusion construct or a *GFP* construct driven by the cauliflower mosaic virus (CaMV) 35 S promoter in *Nicotiana benthamiana*. GFP fluorescence in epidermal cells was recorded at 3 days post infiltration by confocal laser scanning microscopy. **b**, *Agrobacterium*-mediated transient expression of a the 35 S:*Sr43-GFP* with a 35 S:*PLSP2A-RFP* plastid marker construct in *N. benthamiana*. GFP or RFP fluorescence in epidermal cells was recorded at 3 days post infiltration by confocal laser scanning microscopy. **c**, transient expression of a ZmUbi:*GFP-Sr43* fusion construct with ZmUbi:*NLS-mCherry* nuclear marker in wheat mesophyll cells (upper panel) and a ZmUbi:*GFP* as control (lower panel). GFP and mCherry fluorescence were recorded 16–20 hrs post transfection by confocal laser scanning microscopy. A non-transformed cell is indicated with black arrow. The scale bar in all the images is 20 µm. Nuclei (N), cytoplasm (C) and plastids (P) are indicated on the images with white arrows. The sizes of the GFP and Sr43-GFP fusion proteins are estimated at 27 and 126 kDa, respectively. The experiments were repeated in five (Sr43-GFP, panel a), two (GFP, panel b), three (panel b), five (GFP-Sr43) and two (GFP) independent experiments, respectively, with similar results. In the *N. benthamiana* experiments, we infiltrated two leaves and screened more than 100 nuclei per leaf.

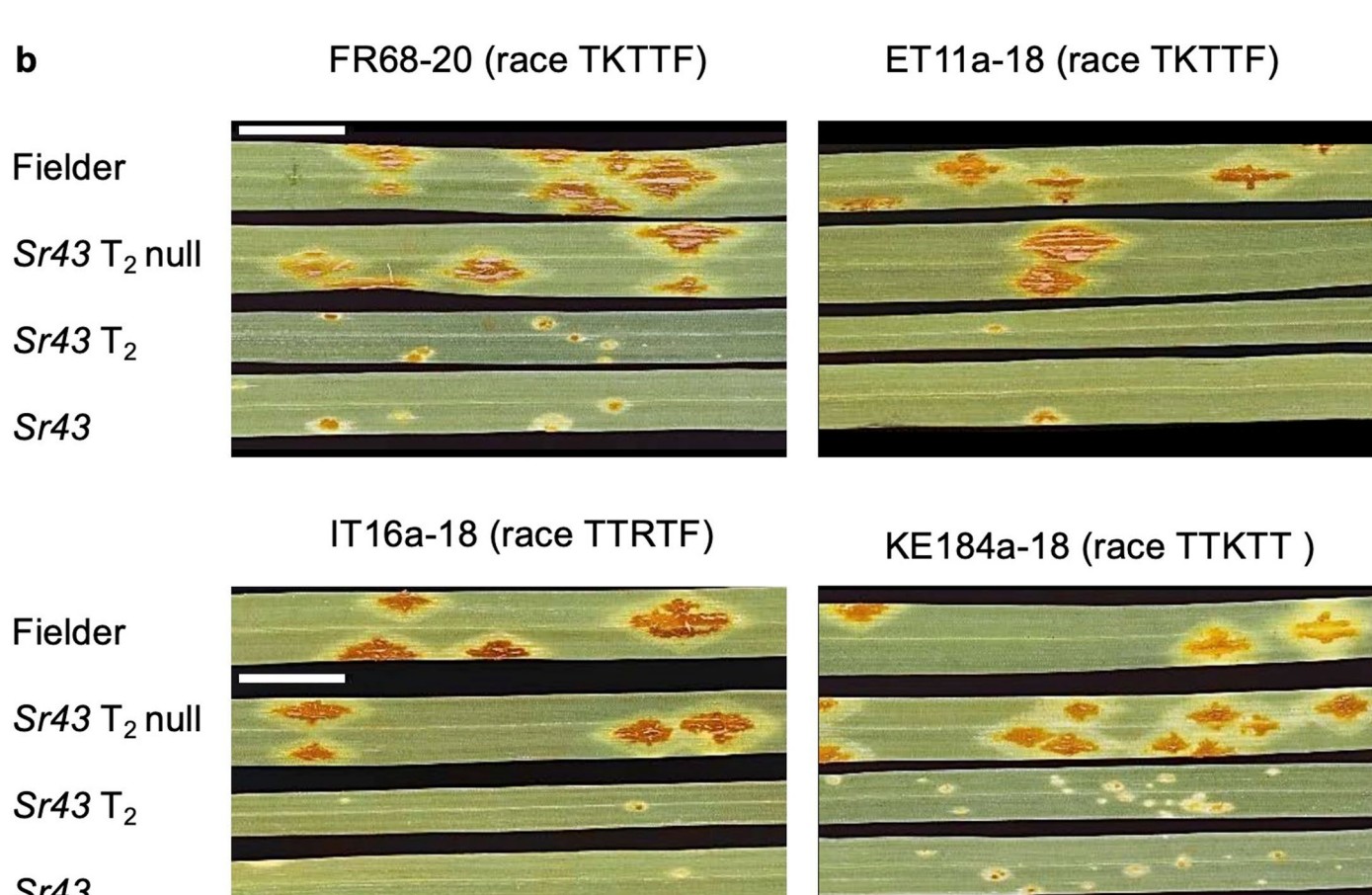

**Extended Data Fig. 4 | Reactions of Sr43 transgenic plants and controls to seven races/isolates of the fungal agent causing stem rust. a**, homozygous transgenic lines from generation $T_1$ alongside the non-transgenic parent cultivar Fielder. **b**, Homozygous transgenic lines and null segregants from generation $T_2$. Scale bar, 1 cm.

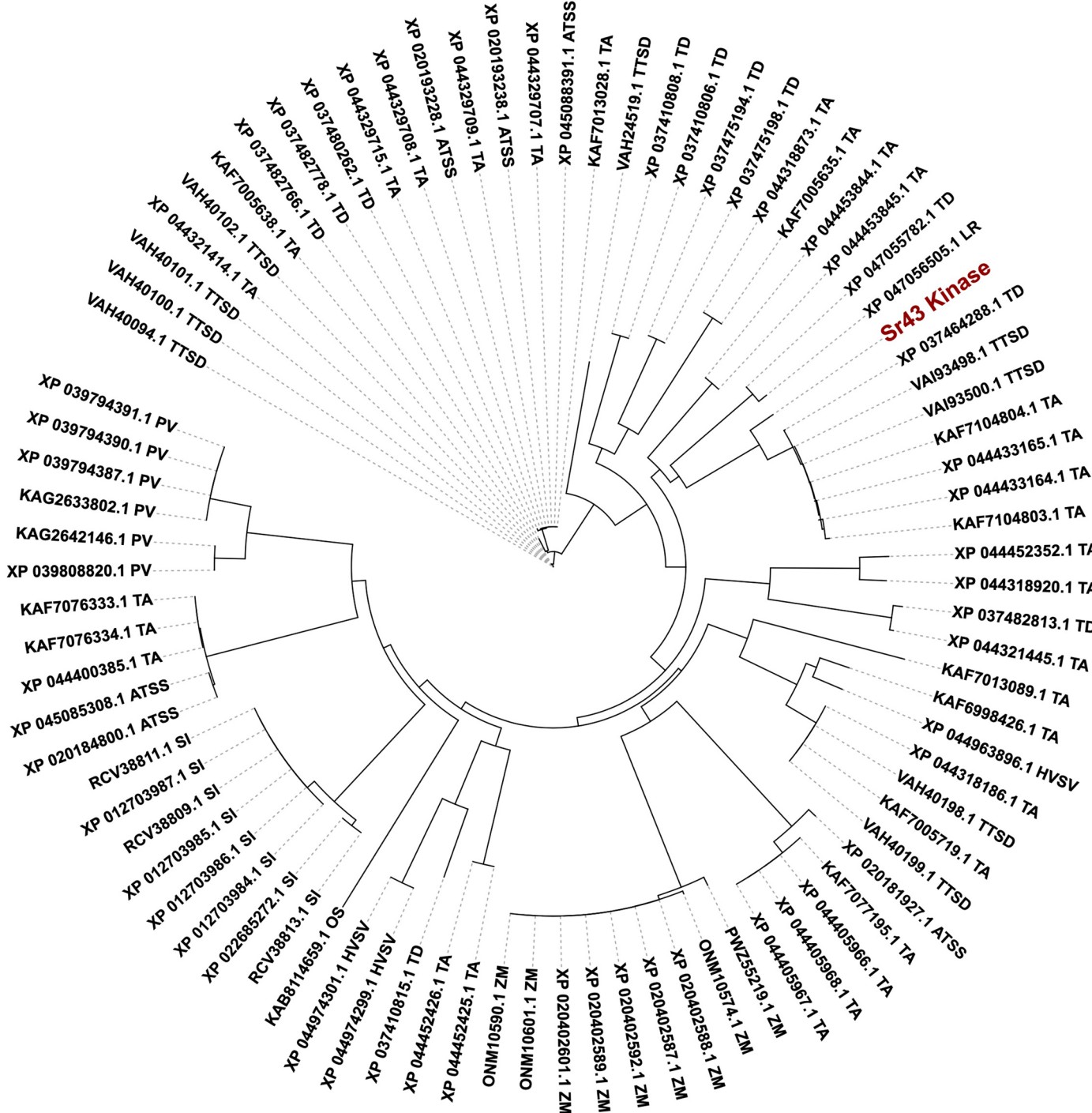

**Extended Data Fig. 5 | Phylogenetic tree of the Sr43 kinase domain with best hits from the NCBI protein database.** A minimum threshold of 94% coverage and an identity of ≥60% were applied. ATSS, *Aegilops tauschii* subsp. *strangulata*; HVSV, *Hordeum vulgare* subsp. *vulgare*; LR, *Lolium rigidum*; OS, *Oryza sativa*; PV, *Panicum virgatum*; SI, *Setaria italica*; TA, *Triticum aestivum*; TD, *Triticum dicoccoides*; TTSD, *Triticum turgidum* subsp. *durum*; ZM, *Zea mays*.

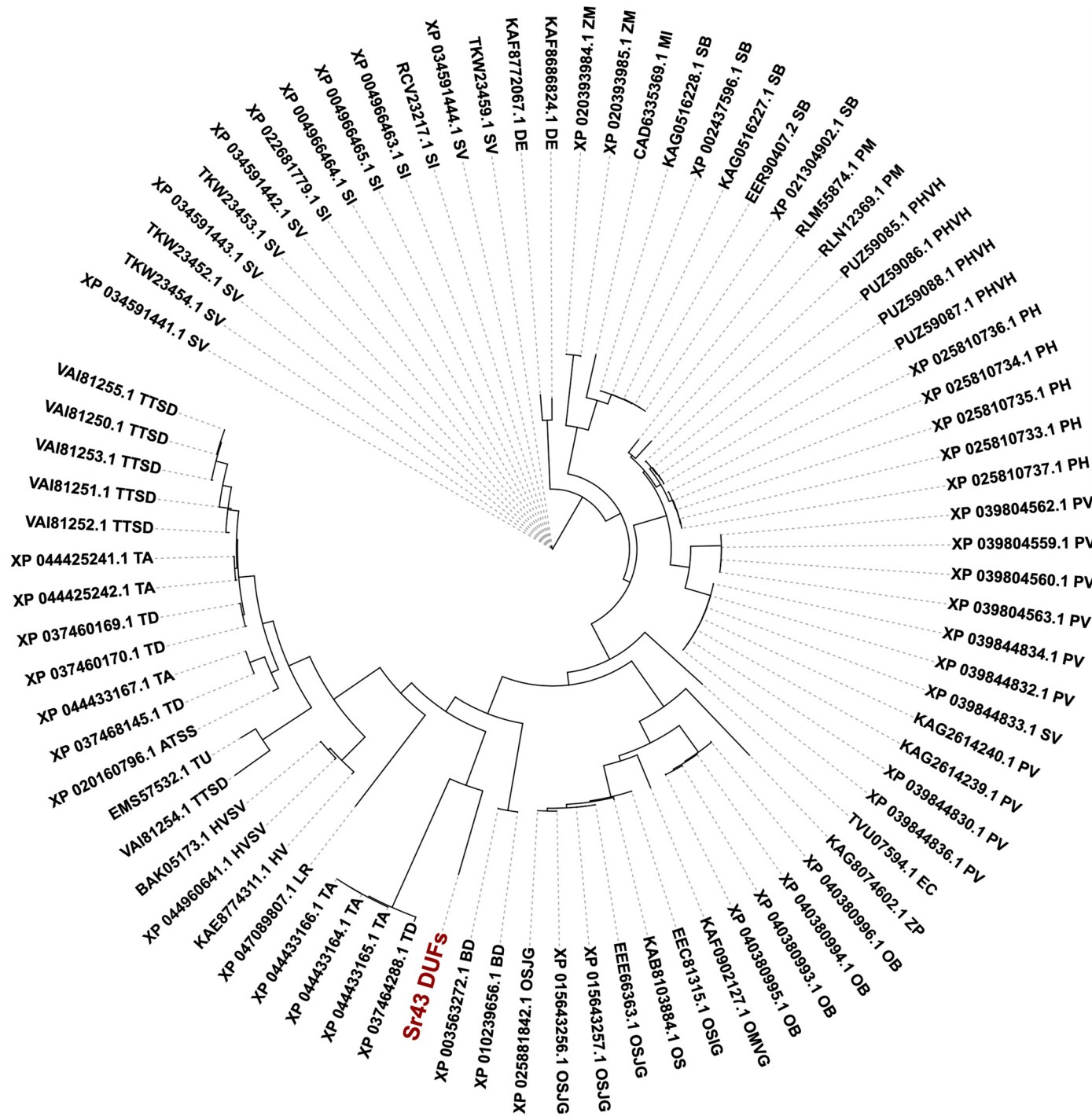

**Extended Data Fig. 6 | Phylogenetic tree of the Sr43 DUF region with best hits from the NCBI protein database.** A minimum threshold of ≥99% coverage and an identity of ≥70% were applied. ATSS, *Aegilops tauschii* subsp. *strangulata*; BD, *Brachypodium distachyon*; DE, *Digitaria exilis*; EC, *Eragrostis curvula*; HV, *Hordeum vulgare*; HVSV, *Hordeum vulgare* subsp. *vulgare*; LR, *Lolium rigidum*; ML, *Miscanthus lutarioriparius*; OB, *Oryza brachyantha*; OMVG, *Oryza meyeriana* var. *granulata*; OS, *Oryza sativa*; OSIG, *Oryza sativa* Indica Group; OSJG, *Oryza sativa* Japonica Group; PH, *Panicum hallii*; PHVH, *Panicum hallii* var. *hallii*; PM, *Panicum miliaceum*; PV, *Panicum virgatum*; SI, *Setaria italica*; SV, *Setaria viridis*; SB, *Sorghum bicolor*; TA, *Triticum aestivum*; TD, *Triticum dicoccoides*; TTSD, *Triticum turgidum* subsp. *durum*; TU, *Triticum urartu*; ZM, *Zea mays*; ZP, *Zizania palustris*.

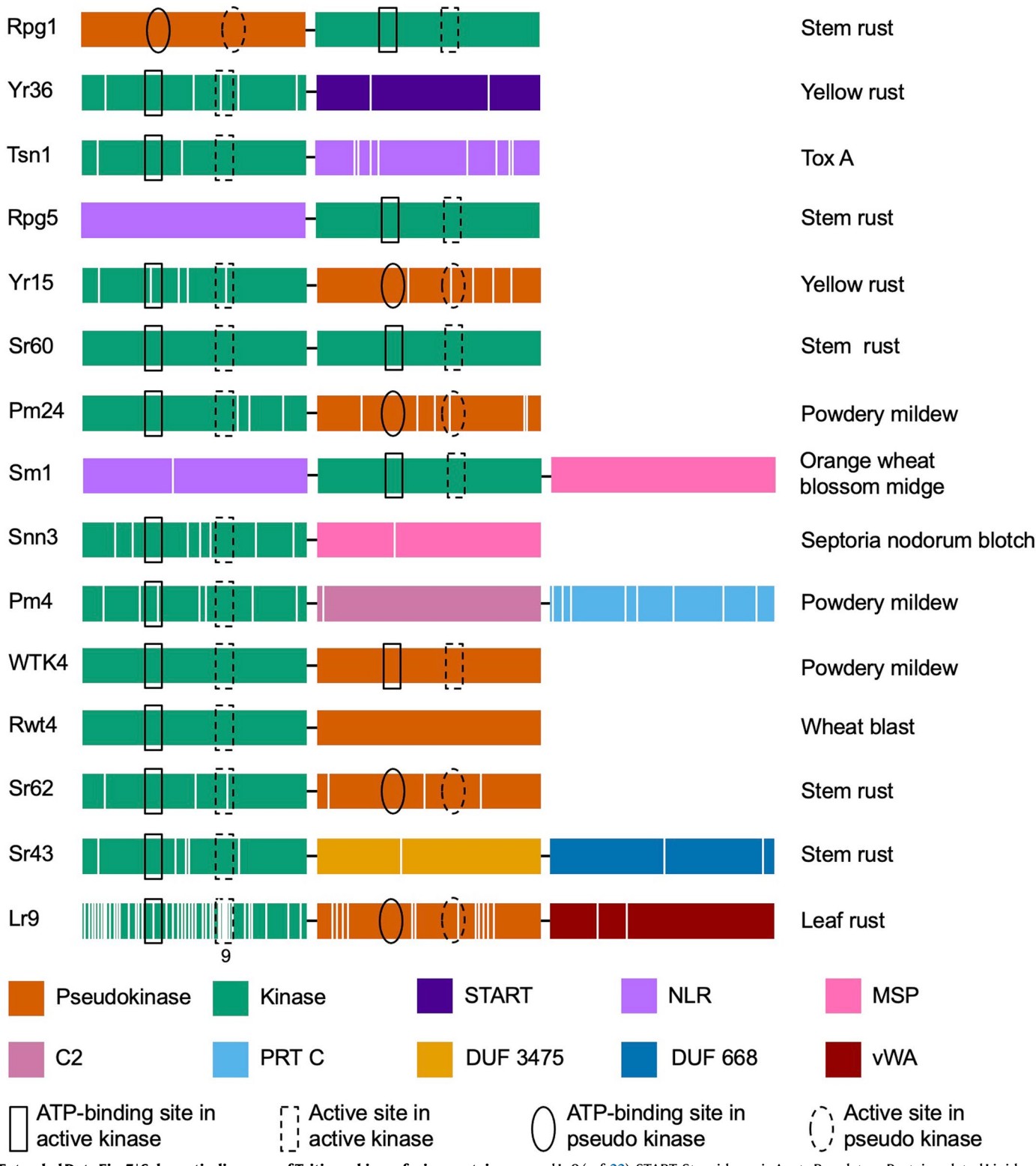

**Extended Data Fig. 7 | Schematic diagrams of Triticeae kinase fusion proteins with disease resistance function.** The ATP-binding and active sites of the kinases are indicated by boxes and circles in the active and pseudo kinases, respectively, as determined by Interpro scanning, homology searching in AlphaFold-derived 3D models and/or BLAST. White bars indicate loss-of-function missense mutations obtained from EMS mutant screens. Eight mutations in the linker domains are not shown. The proteins are organized from top to bottom by date of publication: Rpg1, Yr36, Tsn1, Rpg5, Yr15, Sr60, Pm24, Sm1, Snn3, Pm4 (encoded by splice version 2), WTK4, Rwt4, Sr62 (refs. 9–21), Sr43 (this study)

and Lr9 (ref. 22). START, Steroidogenic Acute Regulatory Protein-related Lipid Transfer; NLR, nucleotide binding and leucine-rich repeat; MSP, major sperm protein; C2, C2 domain; PRT C, phosphoribosyl transferase C-terminal domain; DUF, domain of unknown function; vWA, von Willebrand factor type A domain. White bars indicate the position of EMS-derived loss-of-function mutations (excluding early stop codon mutations) where they are available. The number 9 underneath the active site in the active kinase of Lr9 indicates the presence of nine mutations.

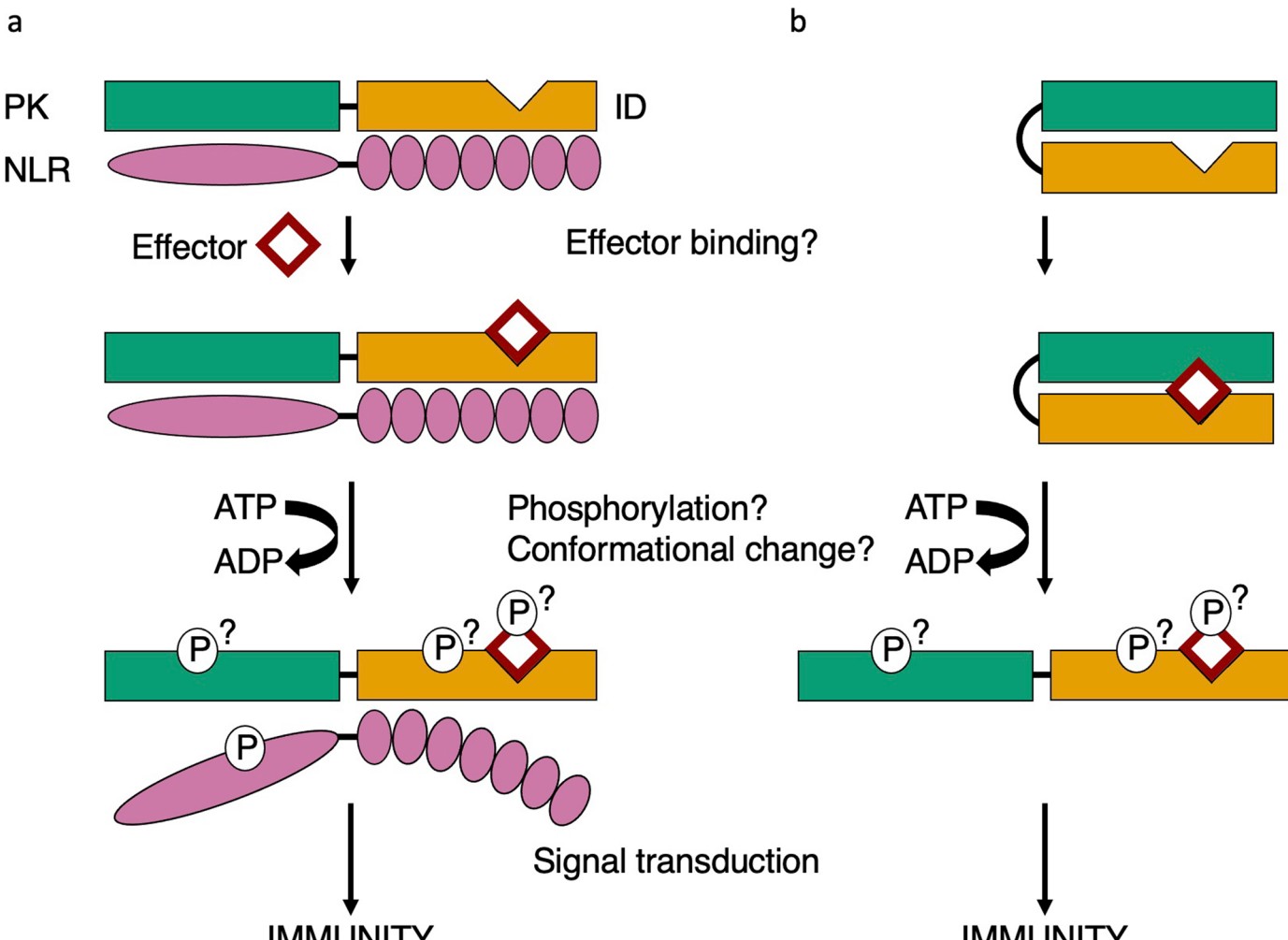

**Extended Data Fig. 8 | Mechanistic model of protein kinase fusion proteins in plant disease resistance.** The protein kinase (PK) fusion protein contains an integrated decoy (ID) that traps the avirulence effector protein (red square). This triggers an autophosphorylation of either the protein kinase or decoy, the effector, or the nucleotide binding leucine-rich repeat (NLR) guard. This results in a conformational change which in turn triggers a signal cascade leading to downstream defense responses and immunity.

| | Brian J. Steffenson |

# Reporting Summary

Nature Research wishes to improve the reproducibility of the work that we publish. This form provides structure for consistency and transparency in reporting. For further information on Nature Research policies, see our Editorial Policies and the Editorial Policy Checklist.

## Statistics

For all statistical analyses, confirm that the following items are present in the figure legend, table legend, main text, or Methods section.

| n/a | Confirmed | |
|---|---|---|
| ☐ | ☒ | The exact sample size (*n*) for each experimental group/condition, given as a discrete number and unit of measurement |
| ☒ | ☐ | A statement on whether measurements were taken from distinct samples or whether the same sample was measured repeatedly |
| ☒ | ☐ | The statistical test(s) used AND whether they are one- or two-sided *Only common tests should be described solely by name; describe more complex techniques in the Methods section.* |
| ☒ | ☐ | A description of all covariates tested |
| ☐ | ☒ | A description of any assumptions or corrections, such as tests of normality and adjustment for multiple comparisons |
| ☒ | ☐ | A full description of the statistical parameters including central tendency (e.g. means) or other basic estimates (e.g. regression coefficient) AND variation (e.g. standard deviation) or associated estimates of uncertainty (e.g. confidence intervals) |
| ☒ | ☐ | For null hypothesis testing, the test statistic (e.g. $F$, $t$, $r$) with confidence intervals, effect sizes, degrees of freedom and $P$ value noted *Give P values as exact values whenever suitable.* |
| ☒ | ☐ | For Bayesian analysis, information on the choice of priors and Markov chain Monte Carlo settings |
| ☒ | ☐ | For hierarchical and complex designs, identification of the appropriate level for tests and full reporting of outcomes |
| ☒ | ☐ | Estimates of effect sizes (e.g. Cohen's *d*, Pearson's *r*), indicating how they were calculated |

*Our web collection on statistics for biologists contains articles on many of the points above.*

## Software and code

Policy information about availability of computer code

| Data collection | No software was used to collect the stem rust phenotype data. |
|---|---|
| Data analysis | Assembly of Sr43 chromosome: Meraculous v 2.0.5 <br><br> Mutant mapping: <br> BWA v. 0.7.12 <br> SAMtools v1.8 <br><br> GBS plotting: <br> bwa mem v0.7.12 (default parameters) <br> SAMtools v.0.1.19 <br> Custom scripts: https://github.com/steuernb/GBS_introgression_line_analysis linked to Zenodo via https://zenodo.org/badge/latestdoi/394326594 <br><br> RNA mapping: <br> Trimmomatic v. <br> Hisat2 v. 2.1.0 <br> Samtools v. 1.8 <br> IGV v. <br><br> Annotation of Sr43: <br> InterPro v. 88.0 |

Myristoylator: Bologna G., et al. N-terminal myristoylation predictions by ensembles of neural networks. Proteomics. 4,1626–1632 (2004).
NLS mapper: Kosugi, S. et al. Systematic identification of yeast cell cycle-dependent nucleocytoplasmic shuttling proteins by prediction of composite motifs. Proc. Natl. Acad. Sci. U.S.A. 106, 10171–10176 (2009).

Phylogenetic tree construction:
https://www.ebi.ac.uk/Tools/msa/clustalo/
https://itol.embl.de/
Protein scan: hmmscan v3.1b2

3D modeling, structure comparison and docking:
AlphaFold v2.0 (https://alphafold.ebi.ac.uk)
Dali (http://ekhidna2.biocenter.helsinki.fi/dali/)
HADDOCK2.4 (https://www.bonvinlab.org/education/HADDOCK-binding-sites/.

For manuscripts utilizing custom algorithms or software that are central to the research but not yet described in published literature, software must be made available to editors and reviewers. We strongly encourage code deposition in a community repository (e.g. GitHub). See the Nature Research guidelines for submitting code & software for further information.

## Data

Policy information about availability of data

All manuscripts must include a data availability statement. This statement should provide the following information, where applicable:

- Accession codes, unique identifiers, or web links for publicly available datasets
- A list of figures that have associated raw data
- A description of any restrictions on data availability

The datasets generated during and/or analyzed in the current study are publicly available as follows. The sequence reads were deposited in the European Nucleotide Archive under project numbers PRJEB52878 (GBS data), PRJEB51958 (chromosome flow sorted data), and PRJEB52088 (RNA-seq data). The Sr43 gene and transcript sequence were deposited in NCBI Genbank under accession number ON237711. The Sr43 chromosome assembly has been deposited in Zenodo with DOI 10.5281/zenodo.6777941. The following public databases/datasets were used in the study: Chinese Spring reference genome39, Gramene (http://www.gramene.org/), https://ensembl.gramene.org/Multi/Tools/Blast, https://wheat.pw.usda.gov/GG3/blast, BLAST non-redundant protein sequence (https://blast.ncbi.nlm.nih.gov/Blast.cgi?PROGRAM=blastx&PAGE_TYPE=BlastSearch&LINK_LOC=blasthome), Taxonomy Browser (https://www.ncbi.nlm.nih.gov/Taxonomy/Browser/wwwtax.cgi?id=1437183), AlphaFold27 (https://alphafold.ebi.ac.uk), Dali28 (http://ekhidna2.biocenter.helsinki.fi/dali/), and HADDOCK25 (https://www.bonvinlab.org/education/HADDOCK-binding-sites/. Source data are provided with this paper.

The following public databases/datasets were used in the study:

Chinese Spring reference genome (IWGSC, 2018)
Gramene: http://www.gramene.org/#
BLAST https://blast.ncbi.nlm.nih.gov/Blast.cgi?PROGRAM=blastx&PAGE_TYPE=BlastSearch&LINK_LOC=blasthome
(non-redundant protein sequence-nr)
Taxonomy Browser: https://www.ncbi.nlm.nih.gov/Taxonomy/Browser/wwwtax.cgi?id=1437183
AlphaFold v2.0 (https://alphafold.ebi.ac.uk)
Dali (http://ekhidna2.biocenter.helsinki.fi/dali/)
HADDOCK 2.4 (https://www.bonvinlab.org/education/HADDOCK-binding-sites/.

Source data are provided with this paper.

# Field-specific reporting

Please select the one below that is the best fit for your research. If you are not sure, read the appropriate sections before making your selection.

☒ Life sciences ☐ Behavioural & social sciences ☐ Ecological, evolutionary & environmental sciences

For a reference copy of the document with all sections, see nature.com/documents/nr-reporting-summary-flat.pdf

# Life sciences study design

All studies must disclose on these points even when the disclosure is negative.

| | |
|---|---|
| Sample size | No sample size calculation was chosen. |
| Data exclusions | We did not exclude any data points from the study. |
| Replication | Mutant and transgenic phenotypes were repeated once with similar outcomes |
| Randomization | The stem rust tests were not randomized. |
| Blinding | No blinding was applied. |

# Reporting for specific materials, systems and methods

We require information from authors about some types of materials, experimental systems and methods used in many studies. Here, indicate whether each material, system or method listed is relevant to your study. If you are not sure if a list item applies to your research, read the appropriate section before selecting a response.

## Materials & experimental systems

| n/a | Involved in the study |
|-----|----------------------|
| ☒ | ☐ Antibodies |
| ☒ | ☐ Eukaryotic cell lines |
| ☒ | ☐ Palaeontology and archaeology |
| ☒ | ☐ Animals and other organisms |
| ☒ | ☐ Human research participants |
| ☒ | ☐ Clinical data |
| ☒ | ☐ Dual use research of concern |

## Methods

| n/a | Involved in the study |
|-----|----------------------|
| ☒ | ☐ ChIP-seq |
| ☒ | ☐ Flow cytometry |
| ☒ | ☐ MRI-based neuroimaging |

