## [Peer Review File · Nature Genetics]

Peer Review Information

Manuscript Title: The wheat stem rust resistance gene Sr43 encodes an unusual protein kinase

Corresponding author name(s): Professor Brande Wulff, Professor Brian (Joel) Steffenson

Reviewer Comments & Decisions:

Decision Letter, initial version:
--

26th Aug 2022

Dear Professor Wulff,

Your Letter, "The wheat stem rust resistance gene Sr43 encodes an unusual protein kinase" has now been seen by 3 referees. You will see from their comments below that while they find your work of interest, some important points are raised. We are interested in the possibility of publishing your study in Nature Genetics, but would like to consider your response to these concerns in the form of a revised manuscript before we make a final decision on publication.

To guide the scope of the revisions, the editors discuss the referee reports in detail within the team with a view to identifying key priorities that should be addressed in revision. In this case, we think all three referees have provided constructive reviews aimed at strengthening the experiments and improving the presentation, and we particularly ask that you address their technical comments as thoroughly as possible with appropriate revisions. We hope that you will find the prioritized set of referee points to be useful when revising your study.

We therefore invite you to revise your manuscript taking into account all reviewer and editor comments. Please highlight all changes in the manuscript text file. At this stage we will need you to upload a copy of the manuscript in MS Word .docx or similar editable format.

*1) Include a "Response to referees" document detailing, point-by-point, how you addressed each

referee comment. If no action was taken to address a point, you must provide a compelling argument. This response will be sent back to the referees along with the revised manuscript.

*2) If you have not done so already please begin to revise your manuscript so that it conforms to our Letter format instructions, available [here](http://www.nature.com/ng/authors/article_types/index.html). Refer also to any guidelines provided in this letter.

[redacted]

We hope to receive your revised manuscript within 3 to 6 months. If you cannot send it within this time, please let us know.

Sincerely,
Wei

Wei Li, PhD
Senior Editor

Nature Genetics
New York, NY 10004, USA
www.nature.com/ng

Reviewers' Comments:

Reviewer #1:

Remarks to the Author:

Yu et al. reported a wheat stem rust resistance gene Sr43, derived from tall wheatgrass (*Thinopyrum elongatum*). The cloning methods are solid, including chromosome flow sorting and sequencing of the wheat-Th. *elongatum* recombinant chromosome 7D in the parental line and eight mutants, and the Sr43 was validated by the transgenic approach. Sr43 is encoding an unusual protein that includes a protein kinase fused to two DUF domains of unknown function (DUF668 and DUF3475), which represent a new form of resistance gene and provides new knowledge on resistance genes with a kinase structure. The authors state that Sr43 DUF668 domain has a predicted protein kinase-like structure. This statement is enforced by the finding of a high-confidence ATP-binding site in DUF668. Therefore, one can conclude that Sr43 has a tandem-kinase protein (TKP)-like structure, similar to previously cloned resistance TKPs, including Lr9 that was submitted back-to-back with the current manuscript describing Sr34. These amazing findings of both Sr43 and Lr9 are justifying the publication of both manuscripts back-to-back in Nature Genetics.

The authors proposed the guard model, similar to some NLRs, while ignoring other possible models. The predicted kinase-like structure for DUF668 is implying that Sr43 is more similar to other Tandem kinases, and therefore fits more the model proposed in the literature for tandem kinase resistance genes (e.g. Klymiuk et al 2021). Therefore, the proposed model for the mode of action of Sr43 needs to be improved, taking into account different scenarios. We recommend this paper for publication in Nature Genetics after a minor revision.

Comments and Suggestions:

1. Line 129-131: This sentence is claiming that "The closest homolog of the Sr43 kinase domain was the serine/threonine kinase interleukin-1 receptor associated kinase (STKc IRAK) (Supplementary Fig. 12), which indicates that the Sr43 kinase is conserved between animals and plants". We find this statement as inconsistent with the statement made in lines 182-185: " We identified 183 proteins harboring either the kinase domain or the two DUFs alone across the Poaceae family spanning 60 million years of evolution (Supplementary Tables 13 and 14)". The stated similarity of Sr43 kinase domain with STKc IRAK is only 30.4% (Supplementary Fig. 12), while the stated similarity of Sr43 with kinase domain alone across the Poaceae family is 63-94% (Table S13 is missing a title. We had to guess that it is comparing kinase domains). Therefore, based on data provided by the authors STKc IRAK is not the closest homolog of the Sr43 kinase domain. Furthermore, we do not think that 30% similarity is enough to state that the Sr43 kinase is conserved between animals and plants.

2. Line 144-146: The authors state that "The domain structure of Sr43 was thus clearly different from that of proteins encoded by the ~283 cloned plant resistance genes, which were largely (78%) extracellular or intracellular immune receptors (Supplementary Table 1)". However, in lines 151-153 the authors state that the Sr43 DUF668 domain has a predicted protein kinase-like structure, enforced by the finding of a high-confidence ATP-binding site in DUF668. Therefore, one can conclude that Sr43

has a tandem-kinase structure, and maybe even kinase-pseudokinase structure similar to several other resistance genes, yet it is more similar to Lr9 since it is fused to DUF3475 which is another domain of unknown function. These amazing findings are justifying the publication of Lr43 and Lr9 back-to-back in Nature Genetics.

3. Line 170-171. Please provide information to justify the use of "Broad-spectrum" resistance of Sr43. How many isolates were tested so far that show avirulence on Sr43? (Including previously published literature). Maybe the term "Wide-spectrum" will fit better, or "conferred high levels of resistance to a wide range of isolates"?

4. Extended Data Fig.8: The model is based on the 20 years old Guard model described for NLRs, which is not necessarily fitting kinase or tandem kinase R-gene. The model and the discussion should include other potential options, such as those previously described for non-NLR R-genes (e.g Sánchez-Martín et al 2021, Current Opinion in Plant Biology, 62, p.102053; Klymiuk et al. 2021. Molecular Plant-Microbe Interactions, 34(10), pp.1094-1102).

5. Supplementary Table 1: Some inaccurate information is found in Supplementary Table 1. For example, Pm17 is originated from rye, and Pm2a was from *Ae. tauschii*. Yr27 seems to be from wheat, not barley. The Yr36 (WKS1) is from Wild Emmer Wheat.

6. The authors proposed that the gene is unique to the Triticeae evolving through a gene fusion event that occurred 6.7 to 11.6 million years ago. However, this manuscript is lacking important information on the variation analysis of the functional allele in natural *Th. elongatum* populations.

Reviewed by Tzion Fahima and Yinghui Li

Reviewer #2:

Remarks to the Author:

Yu et al. reported the cloning of stem rust resistance gene Sr43 originated from wheat wild relative *Thinopyrum elongatum*. Sr43 encodes a protein kinase fused two domains of unknown function. The function of Sr43 was validated by EMS-induced mutants and stable genetic transformation. The finding provides additional information of kinase protein with fused domains conferring disease resistance in Triticeae, especially in wheat. The study is well designed and the results are solid and reliable.

Major comments are:

1. In the manuscript, the authors proposed the kinase-NLR working model for the disease resistance. From the data presented by this manuscript, no NLR candidate was found in the 10 confirmed mutants to support the hypothesis (Extended data Fig. 8). Neither genetic interaction evidence is available also in the current stage. The current available data is focused only on the cloning and functional validation of Sr43. The functions of kinase and fusion domains in the stem rust resistance need further dissection to support the claim that the kinase recognizes the Avr and the phosphorylation of the kinase protein and NLR receptor.
2. Several of the isolated kinase fusion proteins confer broad spectrum resistance. Is there any resistance spectrum information available for the Sr43 since it was introgressed into wheat 45 years ago?

Reviewer #3:

Remarks to the Author:

The study cloned a stem rust resistance gene Sr43, a gene was transferred from a wild grass, in wheat. The candidate gene was discovered through screening EMS mutants and sequencing flow-sorted chromosomes from multiple mutants and the parental line. A resistance gain of a susceptible cultivar through gene transfer confirms the candidate gene is Sr43. The resistance gained is race-specific and temperature-sensitive resistance. The gene structure was determined by performing RNA-seq and sequencing full-length transcripts. The gene was found to contain a kinase domain and two DUFs. Structure analysis indicates that one of DUFs has kinase-like activity. Homology search suggests the gene originated through a fusion and maintained in some grass species. The study provides strong evidence for gene cloning and proposes an interesting decoy model for kinase resistance. Although the support for the decoy model is weak from this study, the manuscript was well written. The known resistance genes from previous work were also well summarized in this manuscript.

Here are comments for consideration:

Three copies of the transgene were inferred in one T0 plant by quantifying the selection marker hygromycin. The reviewer wonders how the copy number was calculated? Was the line with one copy of hygromycin used as the control? Why can the gene itself not be used for the quantification? It is likely some transgenic sites carry a truncated fragment. This is a minor issue. But more detail should help avoid the confusion.

The temperature sensitivity of resistance is not uncommon. Is the sensitivity caused by the change in the fungus or the host response? It is likely caused by the host response. Can the expression of the gene simply explain the phenomenon? The transgene is driven by a native promoter. It might be worth examining the expression of the transgene under different temperatures.

About the emergence of the fusion gene, 75% as the identity cutoff was used for blast search. The reviewer wonders if the minimum identity should be reduced to catch highly polymorphic homologs in distant grass species. At least the lower threshold should be tried.

For the subcellular localization data, should a control (e.g., GFP construct without Sr43) be added to strengthen the conclusion. In particular, the signal on the nucleus is relatively weak and no nucleus localization signal domain is present on the gene.

Author Rebuttal to Initial comments

Reviewers' Comments:

Reviewer #1:

Remarks to the Author:

Yu et al. reported a wheat stem rust resistance gene *Sr43*, derived from tall wheatgrass (*Thinopyrum elongatum*). The cloning methods are solid, including chromosome flow sorting and sequencing of the wheat-*Th. elongatum* recombinant chromosome 7D in the parental line and eight mutants, and the *Sr43*

was validated by the transgenic approach. *Sr43* is encoding an unusual protein that includes a protein kinase fused to two DUF domains of unknown function (DUF668 and DUF3475), which represent a new form of resistance gene and provides new knowledge on resistance genes with a kinase structure. The authors state that *Sr43* DUF668 domain has a predicted protein kinase-like structure. This statement is enforced by the finding of a high-confidence ATP-binding site in DUF668. Therefore, one can conclude that *Sr43* has a tandem-kinase protein (TKP)-like structure, similar to previously cloned resistance TKPs, including *Lr9* that was submitted back-to-back with the current manuscript describing *Sr43*. These amazing findings of both *Sr43* and *Lr9* are justifying the publication of both manuscripts back-to-back in Nature Genetics.

The authors proposed the guard model, similar to some NLRs, while ignoring other possible models. The predicted kinase-like structure for DUF668 is implying that *Sr43* is more similar to other Tandem kinases, and therefore fits more the model proposed in the literature for tandem kinase resistance genes (e.g. Klymiuk et al 2021). Therefore, the proposed model for the mode of action of *Sr43* needs to be improved, taking into account different scenarios. We recommend this paper for publication in Nature Genetics after a minor revision.

Comments and Suggestions:

1. Line 129-131: This sentence is claiming that "The closest homolog of the *Sr43* kinase domain was the serine/threonine kinase interleukin-1 receptor associated kinase (STKc IRAK) (Supplementary Fig. 12), which indicates that the *Sr43* kinase is conserved between animals and plants". We find this statement as inconsistent with the statement made in lines 182-185: "We identified 183 proteins harboring either the kinase domain or the two DUFs alone across the Poaceae family spanning 60 million years of evolution (Supplementary Tables 13 and 14)". The stated similarity of *Sr43* kinase domain with STKc IRAK is only 30.4% (Supplementary Fig. 12), while the stated similarity of *Sr43* with kinase domain alone across the Poaceae family is 63-94% (Table S13 is missing a title. We had to guess that it is comparing kinase domains). Therefore, based on data provided by the authors STKc IRAK is not the closest homolog of the *Sr43* kinase domain. Furthermore, we do not think that 30% similarity is enough to state that the *Sr43* kinase is conserved between animals and plants.

Reply: On reflection, we agree that our statement that the 30% homology between *Sr43* and STKc IRAK indicates that 'the two kinases are conserved between animals and plants' is a weak statement. We have instead changed the sentence to read:

"The closest BLAST homolog of the *Sr43* kinase domain was the serine/threonine kinase interleukin-1 receptor associated kinase (STKc IRAK) (Supplementary Fig. 12), indicating that *Sr43* is likely a kinase".

2. Line 144-146: The authors state that "The domain structure of *Sr43* was thus clearly different from that of proteins encoded by the ~283 cloned plant resistance genes, which were largely (78%) extracellular or intracellular immune receptors (Supplementary Table 1)". However, in lines 151-153 the authors state that the *Sr43* DUF668 domain has a predicted protein kinase-like structure, enforced by the finding of a high-confidence ATP-binding site in DUF668. Therefore, one can conclude that *Sr43* has a tandem-

kinase structure, and maybe even kinase-pseudokinase structure similar to several other resistance genes, yet it is more similar to Lr9 since it is fused to DUF3475 which is another domain of unknown function. These amazing findings are justifying the publication of Lr43 and Lr9 back-to-back in Nature Genetics.

Reply: We thank the reviewers for highlighting this incongruency. We used the protein structure comparison server Dali to search for structural similarities between Sr43 and other proteins in the PDB data bank (<http://ekhidna2.biocenter.helsinki.fi/cgi-bin/sans/sans.cgi>). Dali characterized the DUF668 domain as a “receptor-like protein kinase-like” not due to structural similarity with a kinase (since from Alphafold we know that this is not the case because DUF668 is helical and does not resemble a classic kinase structure), but due to the presence of DUF-like domains in other structures that also contain kinase domains that are still uncharacterized. The Dali output was as follows:

Query header	gene name	Description Estimated PPV, description	Biological process Estimated PPV, GO-id, description	Molecular function Estimated PPV, GO-id, description	Cellular component Estimated PPV, GO-id, description	Inverse ec2go, kegg2go	
S001A QUERY Search		0.91 Receptor-like protein kinase-like 0.88 DUF668 domain-containing protein	0.82 GO:0045927 positive regulation of growth				
			0.78 GO:0040008 regulation of growth				
			0.67 GO:0048518 positive regulation of biological process				
			0.53 GO:0050789 regulation of biological process	0.37 GO:0016301 kinase activity		0.43 GO:0005634 nucleus	
			0.53 GO:0065007 biological regulation	0.36 GO:0016772 transferase activity, transferring phosphorus-containing groups		0.41 GO:0043231 intracellular membrane-bounded organelle	
			0.52 GO:0043434 response to peptide hormone	0.35 GO:0016740 transferase activity		0.41 GO:0043227 membrane-bounded organelle	0.36 EC:2.7.-.- GO:0016772
			0.52 GO:1901652 response to peptide compound	0.33 GO:0003824 catalytic activity		0.39 GO:0043229 intracellular organelle	
			0.50 GO:0010243 response to organonitrogen compound			0.39 GO:0043226 organelle	
			0.50 GO:1901698 response to nitrogen compound			0.37 GO:0005622 intracellular anatomical structure	
			0.49 GO:0009725 response to hormone			0.32 GO:0110165 cellular anatomical entity	

In the revised manuscript, we have changed the wording to make this clear:

“We compared the predicted structure of the Sr43 protein to those in the Protein Data Bank²⁸. This identified structural similarities between DUF668 and some receptor-like protein kinase-like proteins outside of their kinase domains.”

3. Line 170-171. Please provide information to justify the use of "Broad-spectrum" resistance of Sr43. How many isolates were tested so far that show avirulence on Sr43? (Including previously published literature). Maybe the term "Wide-spectrum" will fit better, or "conferred high levels of resistance to a wide range of isolates"?

Reply: We changed the wording from “broad-spectrum” to “wide-spectrum”.

4. Extended Data Fig.8: The model is based on the 20 years old Guard model described for NLRs, which is not necessarily fitting kinase or tandem kinase R-gene. The model and the discussion should include other potential options, such as those previously described for non-NLR R-genes (e.g Sánchez-Martín et al 2021, Current Opinion in Plant Biology, 62, p.102053; Klymiuk et al. 2021. Molecular Plant-Microbe Interactions, 34(10), pp.1094-1102).

Reply: We agree that it would be wise to recognize the alternate model in which Sr43 and other protein-kinase fusion proteins may function without an NLR signaling partner. We thank the reviewers for

pointing this out. In the updated discussion we now recognize this scenario and cite the reviews by Klymiuk and colleagues (2021) and Sánchez-Martín and colleagues (2021):

“Alternatively, Sr43 (and by extrapolation other kinase-fusion resistance proteins)^{35,36} may function without an NLR co-signaling partner (Extended Data Fig. 8b).”

5. Supplementary Table 1: Some inaccurate information is found in Supplementary Table 1. For example, Pm17 is originated from rye, and Pm2a was from *Ae. tauschii*. Yr27 seems to be from wheat, not barley. The Yr36 (WKS1) is from Wild Emmer Wheat.

Reply: We apologize for these mistakes. We have now carefully proofread the table and also added the recently cloned *Pm69*. However, we are of the understanding that the cloned *Pm2* gene is derived from *Triticum aestivum*, see Sánchez-Martín et al. (2016) who cite Pugsley & Carter (1953) [The resistance of twelve varieties of *Triticum vulgare* to *Erysiphe graminis tritici*. Aust J Biol Sci. 6:335–46]. The PDF can be obtained here: <https://www.publish.csiro.au/bi/pdf/BI9530335>.

6. The authors proposed that the gene is unique to the Triticeae evolving through a gene fusion event that occurred 6.7 to 11.6 million years ago. However, this manuscript is lacking important information on the variation analysis of the functional allele in natural *Th. elongatum* populations.

Reply: We agree that the proposed work would be interesting. However, such a study would not change the conclusion on when the *Sr43* gene appeared. Our analysis was based on studying the high-quality assemblies of 21 barley genomes (in which the *Sr43* gene was found to always be absent) and 2 rye genomes (one of which had the *Sr43* configuration). Studying the variation analysis of the functional allele in *Th. elongatum* is a different question, albeit an interesting one. But, we must recognize that this would be a major undertaking requiring the configuration of a diverse panel of *Th. elongatum*, extraction of DNA, sequencing of *Sr43* which comprises ~8 kb of sequence (see Figure 2). Moreover, it will be difficult, almost impossible, to make any firm conclusions about structure-function correlations without transforming functional variant alleles into a susceptible wheat cultivar due to the potential of background resistance in the wild germplasm. The experimental design is also complicated by the pronounced variation in ploidy level in *Th. elongatum*, ranging from 2x to 10x (see Table 2 in Zheng et al. (2014). Characterization of *Thinopyrum* species for wheat stem rust resistance and ploidy level. Crop Sci. 54:2663–2672). This phenomenon poses additional complications in terms of (i) obtaining *bona fide* assemblies of *Sr43* variants, and (ii) attributing *Sr43* variant sequences to function due to genome complexity and the potential effect of ‘genome dilution’. In the recent past, we carried out a structure-function analysis of *Sr22*, another stem rust resistance gene that we cloned. This was a considerable endeavor resulting in its own paper (Hatta et al., 2020. Extensive genetic variation at the *Sr22* wheat stem rust resistance gene locus in the grasses revealed through evolutionary genomics and functional analyses. Mol Plant Microbe Interact. 33:1286-1298). We feel that embarking on such a study for *Sr43* would

require an investment beyond the scope of this study to reach firm conclusions relating *Sr43* sequence variation to function.

Reviewed by Tzion Fahima and Yinghui Li

Reviewer #2:

Remarks to the Author:

Yu et al. reported the cloning of stem rust resistance gene *Sr43* originated from wheat wild relative *Thinopyrum elongatum*. *Sr43* encodes a protein kinase fused two domains of unknown function. The function of *Sr43* was validated by EMS-induced mutants and stable genetic transformation. The finding provides additional information of kinase protein with fused domains conferring disease resistance in Triticeae, especially in wheat. The study is well designed and the results are solid and reliable.

Major comments are:

1. In the manuscript, the authors proposed the kinase-NLR working model for the disease resistance. From the data presented by this manuscript, no NLR candidate was found in the 10 confirmed mutants to support the hypothesis (Extended data Fig. 8). Neither genetic interaction evidence is available also in the current stage. The current available data is focused only on the cloning and functional validation of *Sr43*. The functions of kinase and fusion domains in the stem rust resistance need further dissection to support the claim that the kinase recognizes the Avr and the phosphorylation of the kinase protein and NLR receptor.

Reply: We agree that at this stage the kinase-NLR working model is nothing more than a hypothesis which we introduce *only* in the Discussion of the paper, and with carefully chosen wording, such as “kinase fusion proteins may be pathogenicity targets...”, “Perhaps similarly to... this second domain might be...”. We feel that providing a testable working model adds value to the manuscript and believe it is clear from our wording that we have not made any claims that we have demonstrated this model. Notwithstanding, Reviewer 1 also called for us to recognize an alternate mechanistic scenario, e.g. not involving an NLR. In the updated manuscript, we now also mention this second scenario and depict it in the updated Extended Data Fig. 8.

We also agree that to cast light on the *modus operandi* of *Sr43* will require determining the function of the different domains and their possible interaction with Avr*Sr43*. Towards this long-term goal, we have provided data in the updated manuscript demonstrating that *Sr43* is indeed an active kinase. In brief, we expressed a His6-Maltose binding protein-*Sr43* fusion protein in *E. coli* and showed that the purified *Sr43* protein can phosphorylate maltose binding protein DNA gyrase *in vitro* (see Supplementary Figs. 14-16; Supplementary Table 9).

To make decisive mechanistic conclusions about Sr43, however, will likely require the cloning of AvrSr43 and other possible interactors, which we feel goes beyond the scope of this paper.

2. Several of the isolated kinase fusion proteins confer broad spectrum resistance. Is there any resistance spectrum information available for the Sr43 since it was introgressed into wheat 45 years ago?

Reply: This is indeed a pertinent point. However, although *Sr43* was introgressed into wheat some 45 years ago, the original introgression stock carried at least half a *Thinopyrum* chromosome. It is therefore possible that other background (cryptic) genes on the introgressed segment could contribute to resistance, as seen for example in the case of the wheat-*Aegilops speltoides* *Sr32* line which turned out to also carry the gene *SrAes1t* (Mago et al., 2013; DOI: [10.1007/s00122-013-2184-8](https://doi.org/10.1007/s00122-013-2184-8)). It was only relatively recently that the Sr43 introgressed segment was shortened based on *ph1b*-induced homoeologous recombination (Niu et al. 2014; doi.org/10.1007/s00122-014-2272-4). Niu and colleagues tested their shortened introgression line against Ug99 (from East Africa) and 8 US isolates. Our work supports the broad-spectrum efficacy observed by Niu et al., and extends it to other worldwide and genetically distinct isolates.

Reviewer #3:

Remarks to the Author:

The study cloned a stem rust resistance gene *Sr43*, a gene was transferred from a wild grass, in wheat. The candidate gene was discovered through screening EMS mutants and sequencing flow-sorted chromosomes from multiple mutants and the parental line. A resistance gain of a susceptible cultivar through gene transfer confirms the candidate gene is *Sr43*. The resistance gained is race-specific and temperature-sensitive resistance. The gene structure was determined by performing RNA-seq and sequencing full-length transcripts. The gene was found to contain a kinase domain and two DUFs. Structure analysis indicates that one of DUFs has kinase-like activity. Homology search suggests the gene originated through a fusion and maintained in some grass species. The study provides strong evidence for gene cloning and proposes an interesting decoy model for kinase resistance. Although the support for the decoy model is weak from this study, the manuscript was well written. The known resistance genes from previous work were also well summarized in this manuscript.

Here are comments for consideration:

Three copies of the transgene were inferred in one T0 plant by quantifying the selection marker hygromycin. The reviewer wonders how the copy number was calculated? Was the line with one copy of hygromycin used as the control? Why can the gene itself not be used for the quantification? It is likely some transgenic sites carry a truncated fragment. This is a minor issue. But more detail should help avoid the confusion.

Reply: The Original copy number postulation was carried out by the company iDNA Genetics, Norwich, UK using qPCR against the selectable marker *hygromycin B phosphotransferase* according to the procedure published by Bartlett et al., 2008, which we cite in the manuscript. Postulating transgene copy number from the selectable marker allows iDNA Genetics to process many samples from different customers without having to optimize gene-specific probes. Following the concerns raised by the Reviewer we isolated DNA from a T₃ plant, derived from a non-segregating T₂ family as based on T₁ and T₂ phenotyping. We designed gene specific markers for *Sr43*, *hygromycin B phosphotransferase*, and single-copy, three-copy, and six-copy endogenous control genes. These more extensive qPCR analyses suggest that the stable *Sr43* line contains two copies of *Sr43* and three copies of *hygromycin B phosphotransferase* (see new Supplementary Fig. 17 and Supplementary Tables 13, 14). We have updated the manuscript with this new information.

The temperature sensitivity of resistance is not uncommon. Is the sensitivity caused by the change in the fungus or the host response? It is likely caused by the host response. Can the expression of the gene simply explain the phenomenon? The transgene is driven by a native promoter. It might be worth examining the expression of the transgene under different temperatures.

Reply: We agree with the Reviewer that the mechanism of temperature sensitivity is an enticing question, and having cloned *Sr43*, it now offers potential novel experimental avenues to investigate this phenomenon. However, there are likely a plethora of different mechanisms that can account for the temperature sensitivity. Apart from transcriptional regulation, other regulatory mechanisms at the RNA or protein levels include alternative splicing, nonsense-mediated RNA decay, small RNAs, protein folding, protein compartmentalization, posttranslational modification, interaction with other signaling partners, and so on. In this manuscript, we used the temperature sensitivity of *Sr43* to demonstrate that the transgenic line recapitulated a key phenotype of *Sr43* resistance. However, while the proposed transcriptional experiment is interesting, we feel it is beyond the focus and scope of this paper to open up this “Pandora’s box”. Notwithstanding, we did measure *Sr43* transcription by qPCR at the permissive and non-permissive temperatures and saw a slight increase in expression at the non-permissive temperature. For the curiosity of the reviewer, we show the results below:

Temperature regime	Sr43 Ct	GSR Ct	ΔCt (avg. Sr43 Ct - avg. GSR Ct)	$\Delta \Delta Ct$ (avg. ΔCt - avg. 21°C ΔCt)	Normalized Sr43 amount relative to 21°C $2^{-(\Delta \Delta Ct)}$	
21°C	Ct	Ct				
	22.13	21.34				
	22.34	21.71				
	22.34	21.34				
	22.25	21.30				
	21.89	21.28				
	22.39	21.24				
	Average	22.22	21.37	0.86	0.00	1.0 (0.9-1.2)
Std	0.19	0.17	0.22	0.22		
26°C	21.87	21.43				
	21.10	21.90				
	21.99	22.01				
	21.53	21.46				
	21.27	21.29				
	21.86	21.50				
	Average	21.60	21.60	0.00	-0.86	1.8 (1.3-2.4)
	Std	0.36	0.29	0.44	0.44	

About the emergence of the fusion gene, 75% as the identity cutoff was used for blast search. The reviewer wonders if the minimum identity should be reduced to catch highly polymorphic homologs in distant grass species. At least the lower threshold should be tried.

Reply: In Supplementary Table 15, entitled “The presence of Sr43 homologs in various plant species”, the vast majority of search was done using Ensembl Plants (<https://ensembl.gramene.org/Multi/Tools/Blast>) and GrainGenes nucleotide Databases (<https://wheat.pw.usda.gov/blast/>) with the default parameters/options. Very occasionally, a hit with identity as low as 31% identity was observed but most hits had >65% identity to the Sr43 CDS. The search for Sr43 homologues in *Sitopsis* species was done with local BLAST. We searched again with local blast at 50% identity, the results are the same. No additional homolog of Sr43 with lower identity has been found. We have updated the footnote in Supplementary Table 15 to reflect these nuances.

For the subcellular localization data, should a control (e.g., GFP construct without Sr43) be added to strengthen the conclusion. In particular, the signal on the nucleus is relatively weak and no nucleus localization signal domain is present on the gene.

Reply: We thank the reviewer for the suggestion. We included the following additional controls in the revised version:

- 35S:*GFP* alone.
- Colocalization of Sr43-GFP with the PLSP2A plastid marker fused with the mRFP fluorescent protein.
- Localization in wheat protoplasts.

Data from both systems suggest a nuclear and cytoplasmic localization of Sr43 as evidenced by cytoplasmic strands in Sr43 (Extended data Fig 3). The observed difference in localization between wheat protoplast cells and *Nicotiana* cells (where Sr43-GFP was also found in the plastid) might be the result of using two different plant species. Future studies using a stable transgenic line in which a *sr43* mutant is functionally complemented with a Sr43:reporter fusion driven by the native promoter will provide more insight into the localization of Sr43.

Decision Letter, first revision:

Our ref: NG-LE60498R

31st Jan 2023

Dear Dr. Wulff,

Thank you for submitting your revised manuscript "The wheat stem rust resistance gene Sr43 encodes an unusual protein kinase" (NG-LE60498R). It has now been seen by the original referees and their comments are below. The reviewers find that the paper has improved in revision, and therefore we'll

be happy in principle to publish it in Nature Genetics, pending minor revisions to satisfy the referees' final requests and to comply with our editorial and formatting guidelines.

Sincerely,
Wei

Wei Li, PhD
Senior Editor
Nature Genetics
New York, NY 10004, USA
www.nature.com/ng

Reviewer #1 (Remarks to the Author):

The authors have made great efforts and successfully improved the manuscript. All the questions were answered properly and all matters were resolved point by point. We do not have any further comments.

Reviewer #2 (Remarks to the Author):

The revised Sr43 manuscript made significant improvement based on the reviewers' comments and suggestions. The authors also provided additional information for the phosphorylation ability of the Sr43, indicating it is an active kinase. I was satisfied with the revised version and recommend acceptance of the revised manuscript.

Reviewer #3 (Remarks to the Author):

In the revised manuscript, a GFP construct was added as the control for examining the subcellular localization of Sr43. From the result, the signal in the nucleus appears to be identified in the GFP control. If so, the conclusion about the Sr43 localization on the nucleus cannot be drawn. It is important to correctly interpret the experimental result. Please clarify or correct my interpretation.

About the quantification of transgenes using qPCR, endogenous genes with known copy numbers were used to produce references in the revised manuscript. The PCR efficiency of different DNA targets is influenced by many factors, such as amplicon length, GC content, primer efficiency. It is inappropriate

to use different PCR targets for quantitative inference. I was curious about how the copy of the marker gene was determined. It might not be reasonable to do that based on the qPCR. The copy number also might not be the essential data for the manuscript. However, the experiment design needs to be well justified.

Author Rebuttal, first revision:

Below, we have addressed the two remaining concerns raised by Reviewer 3:

Reviewer #3: In the revised manuscript, a GFP construct was added as the control for examining the subcellular localization of Sr43. From the result, the signal in the nucleus appears to be identified in the GFP control. If so, the conclusion about the Sr43 localization on the nucleus cannot be drawn. It is important to correctly interpret the experimental result. Please clarify or correct my interpretation.

Author reply: We agree with the reviewer that we cannot make a firm conclusion about the localization in the nucleus, hence the wording “likely” in the main text:

“... we established that Sr43 likely localizes to the nucleus, cytoplasm, and plastids, as evidenced by the fluorescence detected from the transient expression of a *Sr43-GFP* (green fluorescent protein) construct in *Nicotiana benthamiana* leaf epidermal cells (Extended data Fig. 3). The nuclear and cytoplasmic localization was confirmed in wheat protoplasts transfected with Sr43-GFP (Extended data Fig. 3).”

We would expect the small GFP protein (~27 kDa) to readily diffuse through the nuclear pore. However, the size of the Sr43-GFP fusion protein (~126 kDa) is bigger than what would be expected to readily diffuse into the nucleus (see e.g. Wang and Brattain (2007). The maximal size of protein to diffuse through the nuclear pore is larger than 60 kDa. FEBS Letters 581(17):3164–3170). Other studies have made similar tentative conclusions about nuclear localization of GFP-fusion proteins in *N. benthamiana* leaves and wheat protoplasts. We list some of these studies in the Table below. Much like Sr43, these fusion proteins also do not appear to have a nuclear localization signal.

Protein	Origin	Localization system	Size (with GFP)	Size determination	References
Rx1	Potato	Nicotiana benthamiana leaves	~145 kDa	Western	Slootweg et al., 2010, The Plant Cell
SNC1	Arabidopsis	Nicotiana benthamiana leaves	~190 kDa	Online calculator	Mang et al., 2012, The Plant Cell
RPS4	Arabidopsis	Nicotiana benthamiana leaves	~165 kDa	Online calculator	Mang et al., 2012, The Plant Cell

We have now updated the revised manuscript by reporting the estimated sizes of the GFP and Sr43-GFP fusion protein in the legend of Extended Data Figure 3: “*The sizes of the GFP and Sr43-GFP fusion proteins are estimated at 27 and 130 kDa, respectively.*”

We hope this additional explanation and tweak to the text addresses the Reviewer’s concern. To make any firm conclusions would require additional experimentation, e.g. fusion to other tags, use of native antibodies, use of native promoter, Western blots to confirm protein size and exclude protein degradation, use of biochemical methods to study localization, etc.

Reviewer #3: About the quantification of transgenes using qPCR, endogenous genes with known copy numbers were used to produce references in the revised manuscript. The PCR efficiency of different DNA targets is influenced by many factors, such as amplicon length, GC content, primer efficiency. It is inappropriate to use different PCR targets for quantitative inference. I was curious about how the copy of the marker gene was determined. It might not be reasonable to do that based on the qPCR. The copy number also might not be the essential data for the manuscript. However, the experiment design needs to be well justified.

Author reply: It is not unusual to use a quantitative PCR-based method (e.g. qPCR or digital PCR) to estimate copy number in primary transgenics. This has the advantage of (i) being cost-effective, (ii) being high throughput, (iii) determining early on which explants are likely true transgenics (rather than escapes), and (iv) giving an estimation of the copy number. The copy number in a set of explants can vary widely from 1 or 2 to many tens of copies. Multi-copy explants (beyond 2 copies) may: (i) complicate identifying genetically stable lines for subsequent analysis, (ii) be more prone to complex insertions with a concomitant higher risk of silencing, and (iii) be more prone to non-native expression patterns which could obfuscate the phenotype. It is therefore useful to use an experimental procedure early on (such as qPCR or digital PCR) that allows weeding out such lines. This is a routine step in our wheat transformation pipeline, as in that of many other labs, and hence it was applied here.

We agree with the Reviewer that the copy number per se is not essential for the manuscript. However, it was important to help identify a genetically stable line for subsequent pathology experiments. The company iDNA Genetics have optimized their copy number estimation using the internal control gene *CONSTANS* (Bartlett et al. 2008, cited 222 times) and used it successfully for many years – we refer to this paper in the Materials and Methods. Following the Reviewer’s initial concern viz-a-viz iDNA’s use of the *hpt* gene as a proxy for the gene of interest (in this case *Sr43*), we repeated the experiment of copy number estimation in our own lab with three independent internal controls for genes with 1, 2 and 3 copies in the wheat genome. We demonstrated in the rebuttal that our results agree with the initial estimation provided by iDNA Genetics. We concur with the Reviewer that PCR

efficiency is influenced by many factors. Please note that in our first rebuttal, we therefore also changed the wording in the materials and methods from “*determined copy number*” to “*estimated copy number*”. We have now also introduced the ‘estimated’ wording in the main text: “...*based on qPCR identified a genetically stable line with an estimated two copies of Sr43*”. We hope the additional context provided above and tweak to the wording in the manuscript addresses the reviewer’s concern.

Final Decision Letter:

18th Apr 2023

Dear Dr. Wulff,

I am delighted to say that your manuscript "The wheat stem rust resistance gene Sr43 encodes an unusual protein kinase" has been accepted for publication in an upcoming issue of Nature Genetics.

Your paper will be published online after we receive your corrections and will appear in print in the next available issue. You can find out your date of online publication by contacting the Nature Press Office (press@nature.com) after sending your e-proof corrections. Now is the time to inform your Public Relations or Press Office about your paper, as they might be interested in promoting its publication. This will allow them time to prepare an accurate and satisfactory press release. Include your manuscript tracking number (NG-LE60498R1) and the name of the journal, which they will need when they contact our Press Office.

Please note that *Nature Genetics* is a Transformative Journal (TJ). Authors may publish their research with us through the traditional subscription access route or make their paper immediately open access through payment of an article-processing charge (APC). Authors will not be required to make a final decision about access to their article until it has been accepted. [Find out more about Transformative Journals](https://www.springernature.com/gp/open-research/transformative-journals)

Authors may need to take specific actions to achieve [compliance with funder and institutional open access mandates](https://www.springernature.com/gp/open-research/funding/policy-compliance-faqs). If your research is supported by a funder that requires immediate open access (e.g. according to [Plan S principles](https://www.springernature.com/gp/open-research/plan-s-compliance)) then you should select the gold OA route, and we will direct you to the compliant route where possible. For authors selecting the subscription publication route, the journal's standard licensing terms will need to be accepted, including [self-archiving-and-license-to-publish](https://www.nature.com/nature-portfolio/editorial-policies/self-archiving-and-license-to-publish). Those licensing terms will supersede any other terms that the author or any third party may assert apply to any version of the manuscript.

Please note that Nature Portfolio offers an immediate open access option only for papers that were first submitted after 1 January, 2021.

An online order form for reprints of your paper is available at <https://www.nature.com/reprints/author-reprints.html>. Please let your coauthors and your institutions' public affairs office know that they are also welcome to order reprints by this

method.

If you have not already done so, we invite you to upload the step-by-step protocols used in this manuscript to the Protocols Exchange, part of our on-line web resource, natureprotocols.com. If you complete the upload by the time you receive your manuscript proofs, we can insert links in your article that lead directly to the protocol details. Your protocol will be made freely available upon publication of your paper. By participating in natureprotocols.com, you are enabling researchers to more readily reproduce or adapt the methodology you use. [Natureprotocols.com](http://natureprotocols.com) is fully searchable, providing your protocols and paper with increased utility and visibility. Please submit your protocol to <https://protocolexchange.researchsquare.com/>. After entering your nature.com username and password you will need to enter your manuscript number (NG-LE60498R1). Further information can be found at <https://www.nature.com/nature-portfolio/editorial-policies/reporting-standards#protocols>

Sincerely,
Wei

Wei Li, PhD
Senior Editor
Nature Genetics
New York, NY 10004, USA
www.nature.com/ng